

# Peroxy Radical Measurements by Ethane - Nitric Oxide Chemical Amplification and Laser-Induced Fluorescence / Fluorescence Assay by Gas Expansion during the IRRONIC field campaign in a Forest in Indiana

Shuvashish Kundu[1*], Benjamin L. Deming[1**], Michelle M. Lew[2], Brandon P. Bottorff[2], Pamela Rickly[3***], Philip S. Stevens[2,3], Sebastien Dusanter[4], Sofia Sklaveniti[3,4], Thierry Leonardis[4], Nadine Locoge[4], and Ezra C. Wood[5]

[1]Department of Chemistry, University of Massachusetts, Amherst MA, 01003, United States
[2]Department of Chemistry, Indiana University, Bloomington IN 47405, United States
[3]School of Public and Environmental Affairs, Indiana University, Bloomington, IN 47405, United States
[4]IMT Lille Douai, Université Lille, Département Sciences de l'Atmosphère et Génie de l'Environnement (SAGE), F-59000 Lille, France
[5] Department of Chemistry, Drexel University, Philadelphia PA, 19104, United States

* now at Momentive Performance Materials, Inc., Tarrytown, NY 10591, United States
** now at Department of Chemistry, University of Colorado, Boulder CO 80309, United States
*** now at Cooperative Institute for Research in Environmental Sciences, University of Colorado, Boulder, CO 80309, USA and Chemical Sciences Division, Earth System Research Laboratory, National Oceanic and Atmospheric Administration, Boulder, CO 80305, USA

*Correspondence to*: Ezra Wood (Ezra.Wood@drexel.edu)



**Abstract.** Peroxy radicals were measured in a mixed deciduous forest atmosphere in Bloomington, Indiana, USA, during the Indiana Radical, Reactivity and Ozone Production Intercomparison (IRRONIC) during the summer of 2015. Total peroxy radicals ($[XO_2] \equiv [HO_2] + \Sigma[RO_2]$) were measured by a newly developed technique involving nitric oxide (NO) – ethane ($C_2H_6$) chemical amplification followed by $NO_2$ detection by cavity attenuated phase shift spectroscopy (hereinafter referred to as ECHAMP). The sum of hydroperoxy radicals ($HO_2$) and a portion of organic peroxy radicals ($[HO_2^*] = [HO_2] + \Sigma\alpha_i[R_iO_2]$, $0<\alpha<1$) was measured by the Indiana University Laser-Induced Fluorescence / Fluorescence Assay by Gas Expansion instrument (LIF-FAGE). Additional collocated measurements include concentrations of NO, $NO_2$, $O_3$, and a wide range of volatile organic compounds (VOCs); and meteorological parameters. $XO_2$ concentrations measured by ECHAMP peaked between 13:00 to 16:00 local time, with campaign average concentrations of $41 \pm 15$ ppt ($1\sigma$) at 14:00. Daytime concentrations of isoprene averaged $3.6 \pm 1.9$ ppb ($1\sigma$) whereas average concentrations of NOx ($[NO] + [NO_2]$) and toluene were 1.2 ppb and 0.1 ppb, respectively, indicating a low impact from anthropogenic emissions at this site.

We compared ambient measurements from both instruments and conducted a calibration source comparison. For the calibration comparison, the ECHAMP instrument, which is primarily calibrated with an acetone photolysis method, sampled the output of the LIF-FAGE calibration source which is based on the water vapor photolysis method and, for these comparisons, generated a 50-50% mixture of $HO_2$ and either butane or isoprene-derived $RO_2$. A bivariate fit of the data yields the relation $[XO_2]_{ECHAMP} = 0.88 ([HO_2]+[RO_2])_{IU\_cal} + 6.6$ ppt, with an $R^2$ of 0.99. This level of agreement is within the combined analytical uncertainties for the two instruments' calibration methods.

A linear fit of all the 30-minute averaged $[XO_2]$ ambient data with the 1-minute averaged $[HO_2^*]$ data (one point per 30 minutes) yields the relation $[XO_2] = 0.85 [HO_2^*] + 3.3$ ppt, with an $R^2$ of 0.67. Day to day variability in the $[XO_2]/[HO_2^*]$ ratio was observed. The lowest $[XO_2]/[HO_2^*]$ ratios between 13:00 and 16:00 were 0.8 on 13 and 18 July, whereas the highest ratios of 1.1 to 1.3 were observed on 24 and 25 July – the same two days on which the highest concentrations of isoprene and ozone were observed. Although the exact composition of the peroxy radicals during IRRONIC is not known, 0-dimensional photochemical modeling of the IRRONIC dataset using the RACM2, RACM2-LIM1, MCM 3.2, and MCM 3.3.1 chemical mechanisms predicts afternoon $[XO_2]/[HO_2^*]$ ratios of between 1.1 to 1.5 depending on mechanism. Differences between the observed ambient $[XO_2]/[HO_2^*]$ ratio and that predicted with the 0-D modeling can be attributed to deficiencies in the model, errors in the two measurement techniques, or both. Although these comparison results are encouraging and demonstrate the viability of using the new ECHAMP technique for field measurements of peroxy radicals, further research investigating the overall accuracy of the measurements and possible interferences from both methods is warranted.

## 1. Introduction

Peroxy radicals in the atmosphere comprise hydroperoxy ($HO_2$) and organic peroxy radicals ($RO_2$, R = organic group). The most important sources of peroxy radicals are the reactions of oxidants (OH, $O_3$, and $NO_3$) with volatile organic



compounds (VOCs), photolysis of oxygenated VOCs, and decomposition of peroxyacetylnitrate (PAN) (Atkinson, 2000). Chemistry involving "ROx" radicals ([ROx] ≡ [OH] + [RO₂] + [HO₂]) leads to the formation of ozone (O₃), oxygenated VOCs, and secondary aerosol particles (Kroll and Seinfeld, 2008;Ng et al., 2008;Atkinson and Arey, 2003;Akimoto, 2016;Claeys et al., 2004). The chemical identity and concentrations of peroxy radicals can provide important information on

atmospheric oxidation processes such as ozone production, the removal efficiency of primary pollutants, and radical budgets. This information is ultimately required to formulate pollution control strategies and to evaluate the impacts of atmospheric chemistry on health and global climate. It is therefore crucial to understand the concentrations and chemistry of $RO_x$ radicals in the atmosphere.

        Comparison of measured radical concentrations to those produced by photochemical models is a common exercise
used to assess our understanding of atmospheric chemistry. Discrepancies of a factor of two or more between measured and modeled OH concentrations have been reported in biogenic VOC-rich forest environments (Lelieveld et al., 2008;Lu et al., 2012;Pugh et al., 2010), suggesting that our knowledge of atmospheric photochemistry is deficient. Similarly, discrepancies between measured and modeled peroxy radicals have suggested the presence of unknown sources or sinks of peroxy radicals (Griffith et al., 2013;Wolfe et al., 2014). These findings have fueled research into the oxidation mechanisms of biogenic
VOCs, especially isoprene (e.g., Wennberg et al., 2018). Although much has been learned in the past decade, the atmospheric fate of biogenic VOCs remains incompletely understood.

        Some past model-measurement comparisons are difficult to interpret because of measurement errors that have recently been discovered. Measurements of OH by the laser-induced fluorescence technique can be affected by a sampling-related interference which can exceed the actual concentration of OH (Mao et al., 2012). Similarly, many previous
measurements of HO₂ by chemical conversion to OH through the HO₂ + NO → OH + NO₂ reaction using both the LIF-FAGE and the perCIMS techniques have been shown to have been affected by a variable contribution from organic peroxy radicals (Fuchs et al., 2011;Hornbrook et al., 2011) and the LIF-based measurements subject to this interference are now referred to as HO₂* ([HO₂*] ≡ [HO₂] + $\alpha_i\Sigma$[R$_i$O₂], 0 < α < 1). The sensitivity of the LIF-FAGE technique to each type of organic peroxy radical varies with the amount of NO added for the conversion and is instrument-dependent but in general is
highest (~90%) for β–hydroxy peroxy radicals derived from alkenes and lowest for those derived from small alkanes (Fuchs et al., 2011;Lew et al., 2018;Whalley et al., 2013).

        Discrepancies between measured and model-predicted OH and XO₂ concentrations can be caused by a combination of measurement errors, missing or incorrect chemistry in models and erroneous model constraints. Measurement errors can be evaluated by the comparison of atmospheric measurements by multiple techniques. Several HOx intercomparison projects
have been conducted in the past few decades (Eisele et al., 2003;Fuchs et al., 2010;Hofzumahaus et al., 1998;Ren et al., 2012;Schlosser et al., 2009). There have been few intercomparisons, however, of total peroxy radical ([HO₂] + ∑[RO₂]) measurements and these have produced mixed results. For example, excellent agreement between the matrix isolation electron spin resonance (MI-ESR) and the RO$_x$ LIF-FAGE techniques was observed in a chamber study involving HO₂, CH₃O₂, and C₄H₇O₂ produced by the oxidation of methane and 1-butene (Fuchs et al., 2009). An earlier comparison of XO₂





measurements between a CO-based chemical amplifier (PERCA) and MI-ESR showed overall agreement of within 10% (Platt et al., 2002). In contrast, $XO_2$ measurements in a forest from two similar CO-based chemical amplifiers differed by more than a factor of three (Burkert et al., 2001). This disagreement was attributed to sampling losses on a rain cover. Similarly, $XO_2$ measurements from two CO-based chemical amplifiers during the airborne African Monsoon

Multidisciplinary Analysis (AMMA) campaign differed by factors of 2-4 when the usual relative humidity-dependent calibration (Mihele and Hastie, 1998) was used for the chemical amplifier data (Andrés-Hernández et al., 2010). As a result, the relative humidity dependence of the chemical amplification technique has been questioned (Andrés-Hernández et al., 2010;Sommariva et al., 2011) despite strong experimental evidence (Butkovskaya et al., 2007;Butkovskaya et al., 2005;Butkovskaya et al., 2009;Mihele et al., 1999;Mihele and Hastie, 1998). Due to the paucity of $XO_2$ measurement

intercomparisons and these new questions regarding the impact of relative humidity on the traditional chemical amplifier technique, further intercomparisons involving different instruments are required before we have enough confidence in the measurements to interpret model-measurement discrepancies as arising from unknown chemistry in models.

This paper presents measurements of $XO_2$ in a mixed deciduous forest by the new Ethane CHemical AMPlifier (ECHAMP) technique (Wood et al., 2017), which is a variation of the traditional chemical amplification or "PERCA"

method (Cantrell and Stedman, 1982;Hastie et al., 1991;Wood and Charest, 2014). Measured $XO_2$ concentrations at this high isoprene, low $NO_x$ environment are described along with supporting measurements of ozone ($O_3$), nitrogen oxides ($NO_x$), biogenic and anthropogenic VOCs, and meteorology. We compare measurements of $XO_2$ by ECHAMP with collocated ambient measurements of $HO_2^*$ by the Indiana University LIF-FAGE technique. We also describe calibration comparison experiments in which ECHAMP, which was calibrated by an acetone photolysis calibration method, quantified radical

concentrations produced by the LIF-FAGE calibration source which is based on the more common water photolysis method. Ozone formation rates are also calculated based on measured $XO_2$ and NO concentrations.

## 2      Experimental Section

### 2.1     Site description

The measurements were carried out at the Indiana University Research and Teaching Preserve (IURTP) field laboratory

during the Indiana Radical, Reactivity and Ozone Production Intercomparison (IRRONIC) campaign over the time period of 9 July – 8 August 2015. The IURTP is located in a mixed deciduous forest 1 km from the perimeter road for Indiana University in Bloomington (Fig. 1). Instrument inlets and related instrumental accessories were set atop a 3 meter scaffolding platform in a clearing behind the IURTP building. The height of the scaffolding was several meters below the forest canopy. The major analytical instruments and gas cylinders were housed inside the building.





## 2.2 ECHAMP Measurements of Total Peroxy Radicals (XO₂)

$XO_2$ concentrations were quantified using a newly developed analytical technique, which involves chemical amplification by ethane ($C_2H_6$) - nitric oxide (NO) followed by nitrogen dioxide ($NO_2$) detection using cavity attenuated phase shift spectroscopy (hereinafter referred as ECHAMP: Ethane CHemical AMPlifier) (Wood et al., 2017). This instrument can be

thought of as a descendent of "traditional" chemical amplifiers, also known as PERCA, in which ambient air is mixed with carbon monoxide and nitric oxide and the resulting "amplified" $NO_2$ measured by the luminol technique (Cantrell and Stedman, 1982;Clemitshaw et al., 1997;Kartal et al., 2010;Mihele and Hastie, 2000). Our initial peroxy radical sensor (Wood and Charest, 2014) relied on the original CO/NO amplification chemistry but utilized a modern, highly sensitive $NO_2$ detection method: cavity attenuated phase shift spectroscopy (CAPS) (Kebabian et al., 2007;Kebabian et al., 2008). The

major modification made for the ECHAMP method used for the measurements described in this study is that ethane ($C_2H_6$) replaces CO as a reagent. This results in greatly improved deployability thanks to the relative safety of $C_2H_6$ compared to CO, a smaller dependence of the sensitivity on relative humidity, but at the expense of lower amplification factors. Details of the experimental technique are described elsewhere (Wood et al., 2017) but its deployment at the IURTP is described here.

The ECHAMP inlet was attached to scaffolding at a height of 3 m. Ambient air was sampled at a flow rate of 5.5 standard

liters per minute (SLPM) into a 0.4 cm inner diameter (ID) glass sampling cross internally coated with halocarbon wax (Halocarbon Products Corp., series 1500) and externally coated with PTFE tape. The sampled air then entered two identical reaction chambers (0.4 cm ID × 61 cm, FEP tubing) at a flowrate of 0.87 SLPM - see schematic in Wood and Charest (2014). The total residence time in the sampling cross before entering the reaction chambers was approximately 18 ms.

At any given point in time, one reaction chamber operated in "amplification" (ROx) mode while the other operated in

"background" (Ox) mode. In "RO$_x$" mode, the air was immediately mixed with NO and $C_2H_6$ in the "upstream" reagent addition port and, 0.1 second later, mixed with nitrogen ($N_2$) in the "downstream" reagent addition port, effecting the following radical propagation reactions:

| | |
|---|---|
| $RO_2 + NO \rightarrow RO + NO_2$ | R1 |
| $RO + O_2 \rightarrow HO_2 + products$ | R2 |
| $HO_2 + NO \rightarrow OH + NO_2$ | R3 |
| $OH + C_2H_6 \rightarrow H_2O + C_2H_5$ | R4 |
| $C_2H_5 + O_2 + M \rightarrow C_2H_5O_2 + M$ | R5 |
| $C_2H_5O_2 + NO \rightarrow C_2H_5O + NO_2$ | R6 |
| $C_2H_5O + O_2 \rightarrow CH_3CHO + HO_2$ | R7 |

Reactions R3 through R7 repeat several times, leading to the formation of $NO_2$ that is subsequently measured by a CAPS sensor. In background (O$_x$) mode, the $N_2$ and $C_2H_6$ flows were switched: sampled air was mixed with NO and $N_2$ upstream and $C_2H_6$ downstream. During this sampling mode, sampled radicals are removed by a combination of reactions R1, R2, R3 and finally the reaction of OH with NO to form HONO. The flowrates of NO, $N_2$ and $C_2H_6$ were each maintained at 45 sccm





using mass flow controllers (MKS model 1179A and Alicat MC series). Cylinder concentrations of NO and $C_2H_6$ (Indiana Oxygen) were 21.1 ppm and 30%, respectively, leading to concentrations in the reaction chamber of 0.9 ppm and 1.4%, respectively. Both upstream and downstream injections were delivered with PFA tubing (0.16 cm i.d. × 6 m). Each reaction chamber alternated between $RO_x$ and $O_x$ mode every 45 seconds on an anti-synchronized schedule using four solenoid valves

controlled by Labview software (National Instruments). After the downstream reagent addition, the air from each reaction chamber flowed through 1 m of 0.32 cm ID FEP tubing, a particulate matter filter (United Filtration Systems, Inc., DIF BN60), and another 6 m of tubing before entering identical CAPS monitors located inside the laboratory. The CAPS $NO_2$ measurements during "$RO_x$" mode are from ambient $NO_2$, $NO_2$ from the reaction of NO and $O_3$ in the reaction chamber and transport tubing, and $NO_2$ from the chemical amplification reactions involving $HO_2$ and $RO_2$ (R1 through R7). In "$O_x$"

mode, the CAPS measures $NO_2$ from the first two categories above and $NO_2$ produced by R1 and R3 but not from the amplification reactions (R3 to R7), as ethane is not added until all radicals are removed by formation of HONO.

The concentrations of peroxy radicals were calculated by dividing the difference between the two CAPS sensors' $NO_2$ measurements ($\Delta NO_2$) between "$RO_x$" and "$O_x$" modes by an experimentally determined amplification factor F:

$$[RO_2] + [HO_2] = \Delta[NO_2]_{(CAPS\ A\ -\ CAPS\ B)}/F \qquad\qquad (1)$$

 The RH-dependent amplification factor F was measured using the acetone photolysis method described by Wood and Charest (2014). Briefly, methyl peroxy ($CH_3O_2$) and peroxyacetyl ($CH_3C(O)OO$) radicals (50 – 400 ppt) were produced by the photolysis of acetone vapor and reacted with excess NO to form $NO_2$ which was quantified using a CAPS $NO_2$ sensor.

The accuracy of this calibration method ultimately depends on the accuracy of the CAPS $NO_2$ measurement (see supplementary information (SI)) and knowledge of the products of the reaction of $CH_3O_2$ and $CH_3C(O)OO$ with NO but does not depend on measurements of actinic flux.

The amplification factor F was measured to be 28 at 0% relative humidity (RH) and decreased to 6 at 90% RH (Wood et al., 2017). The RH was typically between 50 and 75% during the afternoon, corresponding to values of F between

20 and 11. These values are based on laboratory calibrations performed before and after the field project. During the field campaign, we attempted to use a variation on the calibration method described by Wood and Charest (2014). Rather than flow air through the headspace over pure acetone to produce dilute acetone vapor, we instead flowed air through the headspace of dilute (1%) aqueous acetone in an attempt to obviate the need to dilute the resulting acetone vapor (i.e., by reducing the vapor pressure of the acetone per Raoult's Law). Inconsistent calibrations resulted, however, and subsequent

laboratory tests demonstrated that the use of aqueous acetone sometimes produced compounds that absorb blue light and therefore interfered with the CAPS $NO_2$ measurement which is based on absorption of light at 450 nm with a bandpass of 10 nm (full width at half maximum). Because field calibrations were unsuccessful, we have increased the measurement uncertainty accordingly (see below). The acetone vapor photolysis calibration results obtained in the laboratory also agreed



with our prototype $H_2O$ photolysis method as described in Wood et al (2017). Further details on the calibration are described in the SI.

Individual peroxy radicals are not detected with equal sensitivity by ECHAMP due to the formation of organic nitrates and organic nitrites in the reaction chambers:

$RO_2 + NO \rightarrow RO + NO_2$                                                         R8a

$RO_2 + NO + M \rightarrow RONO_2 + M$                                             R8b

$RO + O_2 \rightarrow R'O + HO_2$                                                      R9a

$RO + NO + M \rightarrow RONO + M$                                               R9b

Including a sampling loss term, the sensitivity "α" of ECHAMP to individual organic peroxy radicals relative to that of $HO_2$ can be summarized by Equation 2:

$$\alpha_{RiO2} = S_{RiO2}/S_{HO2} = L_i(1-Y_i)(k_{Ri9a}[O_2]/(k_{Ri9a}[O_2] + k_{Ri9b}[NO])) \tag{2}$$

where $S_{RiO2}/S_{HO2}$ is the sensitivity of ECHAMP to individual $RO_2$ compounds relative to that of $HO_2$, $L_i$ is the fractional sampling transmission of an individual organic peroxy species "$R_iO_2$" through the short inlet into the reaction chambers (relative to that of $HO_2$), Y is the alkyl nitrate yield (Y = R8b/(R8a + R8b)), and the remaining terms in parentheses account for alkyl nitrite (RONO) formation. Alkyl nitrate yields increase with carbon backbone number, from less than 0.1% for $CH_3O_2$ to 8% for isoprene to over 25% for C10 and larger alkyl peroxy radicals (Lockwood et al., 2010;Orlando and Tyndall, 2012). Alkyl nitrite (RONO) formation accounts for less than 4% loss for most organic peroxy radicals and is likely negligible for alkene-derived peroxy radicals due to the rapid decomposition of beta hydroxy alkoxy radicals (Atkinson, 1997), but can sequester a calculated 10% of $CH_3O_2$ (Wood et al., 2017). Sampling losses are limited to the 18 ms transit time in the halocarbon wax–coated sampling cross to the tee in which the NO and $C_2H_6$ are added. Mihele et  al. (1999) measured effective first order wall loss rate constants of 3 to 7 s$^{-1}$ for $HO_2$ onto ¼" OD PFA tubing, depending on RH, and ~0.5 s$^{-1}$ for $CH_3O_2$ and $C_2H_5O_2$. Though this would suggest losses in our inlet of up to 12% for $HO_2$ and 1% for the alkyl peroxy radicals, laboratory tests on our inlet have demonstrated losses of less than 2% for $HO_2$ in our inlet and loss rate constants onto various fluoropolymers much lower than presented in Mihele et al. (1999) as described in the SI.

At an RH of 50%, the theoretical 1σ precision of the ECHAMP measurements, limited by only the precision of the CAPS $NO_2$ measurements and the amplification factor, was 0.8 ppt for a 90-second average. The atmospheric variability of $O_3$, which after reaction with NO accounts for most of the $NO_2$ observed by the CAPS sensors, led to an additional contribution to the noise due to the slightly different time responses of the two CAPS sensors. The observed precision during sampling was typically 2.5 ppt (1σ) for 90-second averaging (Wood et al., 2017), leading to a detection limit of 5 ppt for 90-second averaging and 1.6 ppt for 15 minute averages at a signal-to-noise ratio of two. At night, although variability of $O_3$




was negligible, high RH values of over 95% and the resulting low values of F led to detection limits of between 2 ppt and 8 ppt for 90 second average measurements.

We assign an uncertainty of 27% (2σ) to the ECHAMP measurements during the IRRONIC project, comprising the uncertainty in the $NO_2$ calibration of the CAPS sensors (5%), the uncertainty in the relative humidity - dependent amplification factor (usually 16%, but increased to 25% because post-deployment laboratory calibrations were used instead of the unsuccessful field calibrations using aqueous acetone), and the variable sensitivity to speciated peroxy radicals. These uncertainties are more fully described in Wood et al. (2017). Except where noted otherwise, all ECHAMP $XO_2$ measurements presented are 15-minute averages.

## 2.3 Laser-Induced Fluorescence Measurements HO₂*

$HO_2^*$ was measured by the Laser-Induced Fluorescence / Fluorescence Assay by Gas Expansion (LIF-FAGE) technique described in detail elsewhere (Griffith et al., 2013a;Dusanter et al., 2009). Briefly, air is sampled through a pinhole into a low pressure chamber and mixed with NO which converts $HO_2$ into OH. OH radicals are excited by 308 nm radiation from a tunable dye laser and the subsequent fluorescence detected with a time-gated microchannel plate photomultiplier (MCP-PMT) detector. Some organic peroxy radicals are also converted into OH in the LIF-FAGE instrument. Based on laboratory tests, the sensitivities "α" of the LIF-FAGE measurement for the added NO concentrations used in this study relative to $HO_2$ for the following $RO_2$ radicals are 83% for isoprene-$RO_2$, 91% for methyl vinyl ketone $RO_2$, 54% for methacrolein $RO_2$, 65% for ethene-$RO_2$, 65% for toluene-$RO_2$, 15% for propane-$RO_2$, and 31% for butane-$RO_2$ (Lew et al., 2018). The conversion efficiency for other major $RO_2$ can be either estimated or measured to be 5% for $CH_3O_2$ and the acetyl peroxy radical ($CH_3C(O)O_2$), 8% for ethyl peroxy radical ($C_2H_5O_2$), and 31-55% for $RO_2$ compounds from the OH oxidation of high-molecular-weight hydrocarbons (Fuchs et al., 2011;Griffith et al., 2013). These conversion efficiencies are average values weighted over the distribution of isomers where applicable.

The LIF-FAGE instrument was calibrated using a portable calibrator in which quantified amounts of OH/$HO_2$ and $RO_2$ were produced through the photolysis of water vapor by a low-pressure mercury lamp at 184.9 nm (Dusanter et al., 2008). Humid air containing either isoprene (80 ppb) or n-butane (1.4 ppm) entered the rectangular calibrator (1.27 × 1.27 × 30 cm). Light from a low-pressure mercury lamp (UVP Inc, model 11sc1) illuminated a ~3 cm³ photolysis volume through a quartz window. The flow rate of air was maintained at 45 SLPM. A mixture with equal concentrations of $HO_2$ and either $C_5H_8(OH)O_2$ (from isoprene) or $C_4H_9O_2$ (from butane) were produced when isoprene or butane were added to the calibration gas upstream of the photolysis region, respectively. Ozone actinometry was used to quantify the product of the actinic flux and the exposure time ("Ft") in the calibrator (Dusanter et al., 2008). Concentrations of generated peroxy radicals are calculated by the following equation :



$$[HO_2] + [RO_2] = \frac{[O_3][H_2O]\sigma_{H_2O}\varphi_{H_2O}}{[O_2]\sigma_{O_2}\varphi_{O_2}}$$
(3)

where $[O_3]$ is the concentration of ozone generated by the photolysis of $O_2$; $\sigma_{H_2O}$ and $\sigma_{O_2}$ are the absorption cross sections of $H_2O$ and $O_2$ at 184.9 nm, respectively; and $\varphi_{H_2O}$ and $\varphi_{O_2}$ are the photolysis quantum yields, both equal to two (Washida et al., 1971). A value of $7.14 \times 10^{-20}$ cm$^2$ molecule$^{-1}$ (base e) was used for $\sigma_{H_2O}$ (Cantrell et al., 1997;Hofzumahaus et al., 1997). The effective value of $\sigma_{O_2}$ depends on the $O_2$ optical depth and the operating conditions of the mercury lamp and was determined to be $1.20 \times 10^{-20}$ cm$^2$ molecule$^{-1}$ (Dusanter et al., 2008;Lanzendorf et al., 1997). The water vapor mixing ratio was measured by IR absorption spectrometry using a LI-COR 6262 monitor. Ordinarily the ozone mixing ratio is determined using a calibrated photodiode installed in the calibrator (Griffith et al., 2013). The conversion factor (calibration) that converts the photodiode reading to an $O_3$ mixing ratio is determined from separate experiments in which a range of $O_3$ concentrations produced by the calibrator are measured with a UV-absorption $O_3$ sensor. For this project, $[O_3]$ was instead quantified by the ECHAMP CAPS $NO_2$ sensors after conversion to $NO_2$ by reaction with excess NO. This produces a very precise measurement of the sum of $[O_3]$ and $[NO_2]$ (1$\sigma$ precision of 22 ppt for 10 second averages). The accuracy of this ozone determination is thus ultimately traceable to the CAPS $NO_2$ calibration (see SI). Linking the IU FAGE $HO_2^*$ calibration to the ECHAMP $NO_2$ measurement has ramifications for the intercomparison of the IU calibration source and the ambient measurements as discussed in the relevant sections below.

The sensitivity of the instrument is corrected for fluorescence quenching by water vapor. This amounted to a correction of approximately 20% at a water mixing ratio of 1%. The limit of detection of $HO_2^*$ was 0.8 ppt (30 s average, signal-to-noise ratio of two). The overall accuracy of the $HO_2^*$ measurements was ±36% (2$\sigma$). On all days except 22 July, $HO_2^*$ data were collected for 1 minute every 30 minutes and OH was measured during the rest of the 30 minute cycle. On 22 July, OH was not measured and instead the FAGE instrument measured $HO_2^*$ continuously.

## 2.4 Supporting Measurements

Ambient $NO_2$ was measured using a separate CAPS monitor (Aerodyne Research) (Kebabian et al., 2007;Kebabian et al., 2008). The standard 450 nm bandpass filter used by the CAPS monitor was replaced with a 470 nm bandpass filter to eliminate any interference by glyoxal and methyl glyoxal (Kebabian et al., 2008). This reduced the sensitivity by approximately a factor of three but still provided high signal-to-noise ratios (>100) for the ambient measurements. $O_3$ was measured with a UV absorbance monitor (2B Technologies model 202). NO was measured using a Thermo Fisher chemiluminescence sensor (Model 42i Trace Level). NO, $NO_2$, and $O_3$ data were averaged to 1 minute. Additional details regarding the calibrations and baseline measurements for the NO, $NO_2$, and $O_3$ measurements can be found in the SI.

A wide variety of biogenic and anthropogenic VOCs including isoprene and its oxidation products (methyl vinyl ketone and methacrolein), monoterpenes, non-methane hydrocarbons ($C_2$-$C_5$ and $C_6$-$C_{12}$), including aromatics, and



oxygenated VOCs (alcohols, aldehydes and ketones) were measured during IRRONIC. An online GC-FID-FID was used to measure 57 NMHCs (Badol et al., 2004). Ambient air was sampled through a NAFION membrane and NMHCs were trapped at a temperature of -30 °C inside a quartz tube filled with Carbosieve SIII and Carbopack B. A thermodesorption unit (Perkin Elmer, ATD 400) was used to inject the sample into two columns (PLOT alumine and CPSil 5CB) to separate $C_2$–$C_6$ and $C_6$–$C_{12}$ compounds. Two FID detections provided limits of detection of 10–60 pptv at a time resolution of 90 min. A second online GC-FID instrument was used to measure ethanol, isopropanol, methylethylketone and a few monoterpenes (α-pinene, 3-carene) (Roukos et al., 2009). A sampler unit (Markes International, air server Unity 1) allowed continuous sampling of ambient air through a trap held at 12 °C and filled with Carbopack B and Carbopack X. After thermodesorption, the GC separation was performed using a high-polarity CP-Lowox column (Varian, France). Limits of detection reached with this instrument were in the range 10–90 pptv for a time resolution of 90 min. Offline sampling was performed on multisorbent cartridges to measure > C9 anthropogenic compounds (alkanes and aromatics) and monoterpenes (pinenes, terpinenes, limonene, ocimene, terpinolene, camphene, myrcene, borneol, camphor, cumene), and on DNPH (DiNitroPhenylHydrazine) cartridges to measure carbonyls, including formaldehyde, acetaldehyde and higher compounds. The cartridge measurements were integrated over 2-h sampling periods. Technical details can be found in (Ait-Helal et al., 2014;Detournay et al., 2011;Detournay et al., 2013).

Zero-dimensional photochemical modeling of this field campaign data was performed using the Framework for 0-Dimensional Atmospheric Modeling (F0AM) which was constrained by the 30 minute average mixing ratios of the supporting measurements (Wolfe et al., 2016). F0AM was executed using four different chemical mechanisms: two versions of the Regional Atmospheric Chemistry Mechanism (RACM2 and RACM2-LIM1) and the Master Chemical Mechanism (MCM 3.2 and 3.3.1). RACM2 groups various compounds based on similar rates of reaction resulting in 363 reactions from 17 stable inorganics, 4 inorganic intermediates, 55 stable organics, and 43 intermediate organics (Goliff et al., 2013). RACM2-LIM1 incorporates the revision to the isoprene oxidation mechanism (Peeters et al., 2009) that includes the Leuven Isoprene Mechanism (LIM) including a 1,6 H-shift and a 1,5 H-shift for isoprene peroxy radicals. MCM is a near-explicit chemical reaction model resulting in approximately 17000 reactions from 6700 radical species from methane and 142 non-methane species. Similar to the LIM1 mechanism, MCM 3.3.1 was updated to include revisions to the isoprene oxidation mechanism resulting in HOx recycling from peroxy radical H-shift isomerization as well as NOx recycling and updated ozonolysis rate constants.

## 3        Results and Discussion

### 3.1 Calibration comparisons between ECHAMP and IU calibration source

On 24 and 26 July the IU calibration source was positioned so that its output overflowed the ECHAMP inlet. Figure 2 compares the response of ECHAMP to variable concentrations of peroxy radicals generated by the IU calibrator. Concentrations of peroxy radicals were varied by adjusting the mixing ratio of water or by changing the intensity of the UV



lamp. $H_2O$ mixing ratios varied from 0.1 to 1.4%, corresponding to relative humidities between 5 and 45% and F values between 28 and 17. A bivariate fit between the ECHAMP measurements and the concentrations calculated by eq. 1 results in the relation ECHAMP = $(0.88 \pm 0.02) \times$ (IU cal source) + $(6.6 \pm 4.5)$ ppt with an $R^2$ of 0.99. If both instrument's calibrations were perfectly accurate, however, the slope would not be expected to equal unity because the two instrument's

calibration methods do not produce the same type of peroxy radicals. ECHAMP is calibrated with the acetone photolysis method, which produces an equimolar mixture of $CH_3O_2$ and $CH_3C(O)O_2$ radicals (Wood and Charest, 2014). Because a calculated 10% of both of these radicals will be converted to $CH_3ONO$ in the reaction chambers and not be detected, ECHAMP is expected to be 11% (1/0.9) more sensitive to $HO_2$ than to $CH_3O_2$ and $CH_3C(O)O_2$. Moreover, ECHAMP is expected to be between 7 to 12% less sensitive to $RO_2$ from butane and isoprene than to $HO_2$ because of the respective alkyl

nitrate yields for both peroxy radicals: 8% for butane and 7 − 12% for isoprene (Atkinson et al., 1982;Lockwood et al., 2010;Patchen et al., 2007;Paulot et al., 2009). Thus if both instruments' calibrations were perfectly accurate, then the expected slope for the calibration comparison using butane (i.e., 50% $HO_2$ and 50% $C_4H_9O_2$) would be 1.07 (i.e., 1.11 × 0.96) and the expected slope when using isoprene would be between 1.07 and 1.04 depending on the isoprene alkyl nitrate yield. These values differ from the observed slope of 0.88 by 18 to 22%.

15       The 2σ analytical uncertainty for the IU calibration source and ECHAMP measurements are 36% and 27%, respectively. Because the IU calibration source's $O_3$ mixing ratios were determined by ECHAMP, however, a portion of these two uncertainties is correlated. The uncertainty bars in Fig. 2 have been reduced to remove this component of the uncertainty - to 23% for IU (Dusanter et al., 2008) and 26.6% for ECHAMP. The 18 to 22% difference between the observed slope of 0.88 and the expected slope of 1.04 to 1.07 is within the adjusted uncertainties of both the ECHAMP measurements and the IU calibration source. Moreover, that ECHAMP evidently has near identical sensitivity to these two types of organic

peroxy radicals demonstrates that differences in the mechanisms for converting $RO_2$ to $HO_2$ between β-hydroxy and alkyl peroxy radicals do not appear to affect their detection by ECHAMP.

         The excellent linearity of Fig. 2 is notable because the calibrations were performed over a range of relative humidity values, each of which requires a different amplification factor to be used by ECHAMP. If the RH-dependence of the

ECHAMP calibration had been ignored and only the dry calibration factor been used instead, the comparison would have been inferior as indicated by the squares in Fig. 2, for which a linear fit (not shown) gives the relation ECHAMP = 0.69(IU cal source) + 10.8 ppt. This serves as evidence that RH-dependent calibrations are indeed needed for producing accurate results from chemical amplifiers, including traditional CO and NO-based instruments (e.g., PERCA).

**3.2 Ambient concentrations of total peroxy ($XO_2$) radicals, trace gases, and meteorological parameters**

Ambient concentrations (15-minute averages) of $XO_2$, isoprene, ethene, $O_3$, NO, and $NO_2$, along with meteorological parameters are shown in Fig. 3.





The 15-min average $XO_2$ concentrations in the daytime ranged from below the detection limit of ~5 ppt to 77 ppt. Among the VOCs measured, the daytime concentrations of low-molecular weight total alkanes ($C_2$-$C_5$) were the highest (average mixing ratio ± 1 standard deviation: 5.7 ± 3.9 ppb) followed by isoprene (3.6 ± 1.9 ppb), total C2-C5 alkenes (1.1 ± 0.3 ppb), high-molecular-weight alkanes ($C_6$-$C_{14}$, 0.3 ± 0.2 ppb), toluene (0.1 ± 0.1 ppb) and monoterpenes (0.1 ppb). NO concentrations typically peaked at 0.2 to 0.8 ppb between 09:00 - 11:00 and were almost always below 0.2 ppb between 12:00 and 21:00, whereas $NO_2$ concentrations in the daytime ranged between 0.3 to 3 ppb. $O_3$ concentrations varied between 0 to 71 ppb (av. 35.0 ± 8.4 ppb).

Measured $XO_2$ concentrations during IRRONIC exhibited a diurnal profile characterized by low mixing ratios (often below detection limit) between 0:00 – 07:00, increasing values from 07:00 to 13:00, peak values between 13:00 and 16:00, followed by a decrease in the late afternoon, similar to past measurements in other forests (Burkert et al., 2001;Hewitt et al., 2010;Mihele and Hastie, 2003). $XO_2$ mixing ratios were generally positively correlated with concentrations of isoprene, total alkenes, and ozone (Fig. 3). The highest $XO_2$ concentrations of over 60 ppt were measured during the afternoon of 24 and 25 July, coinciding with the highest average concentrations of isoprene (4.4 ppb), total alkenes (1.8 ppb), and $O_3$ (61 ppb), and the lowest average concentration of NO (0.1 ppb). The lowest daytime concentrations of $XO_2$ were observed on 13 July and 15 July, which were also characterized by lower isoprene and ozone mixing ratios and higher $NO_2$ mixing ratios.

We compare our $XO_2$ concentrations with reported $XO_2$ and $HO_2^*$ concentrations from other forests. The observed daytime $XO_2$ mixing ratios (campaign daytime average 26 ppt) at the IRRONIC site at Indiana are similar to those reported in a tropical rain forest in Malaysia (range 2-68 ppt) (Hewitt et al., 2010), in a northern Michigan forest during several intensive campaigns (range 8-65 ppt) (Griffith et al., 2013;Mihele and Hastie, 2003), and in a tropical forest over South America (campaign av. 42 ppt) (Lelieveld et al., 2008). $XO_2$ concentrations at Indiana never exceeded 80 ppt, in contrast to studies in which measured peroxy radical mixing ratios sometimes exceeded 150 ppt (Burkert et al., 2001;Wolfe et al., 2014).

Measurements of peroxy radical and NO concentrations enable ozone production rates to be calculated directly rather than rely on photochemical models. Using the measured concentrations of peroxy radicals and NO, calculated ozone production rates at the IURTP were at most 9 ppb/hr and described more in the SI.

### 3.3 Comparisons of Ambient Peroxy Radical Mixing Ratios

Figure 4 compares ambient [$XO_2$] measurements by ECHAMP (30-minute averages) with the [$HO_2^*$] measurements by LIF-FAGE (1-minute average every 30 minutes) during 13-25 July. Only data from days in which both instruments were operational are shown. Although in general it is preferable to compare measurements with equal time averaging, the precision of ECHAMP during this campaign – typically 2.5 ppt (1σ) for the 1.5 minute average measurements – necessitated this averaging. The diurnal profiles of both measurement sets, divided into 30-minute bins, are displayed in



Fig. 5. Both figures indicate that the ECHAMP and LIF-FAGE measurements are in general well correlated and follow the same diurnal trend, though closer inspection reveals significant day to day and even hour to hour variability in the ratio.

A linear regression of all the $XO_2$ and $HO_2^*$ measurements yields the relationship $[XO_2] = 0.85\,[HO_2^*] + 3.3$ ppt, $R^2 = 0.67$ (Fig 6.). Adjusting this for the calibration difference described in section 3.1, i.e. increasing the $XO_2$ values by a factor of 1.2 yields a slope of 1.02. If the y-intercept is constrained to zero, which is not necessarily appropriate, the slope increases to 0.96, or 1.15 after adjusting for the calibration difference. The individual days with the highest $[XO_2]/[HO_2^*]$ slopes were 24 and 25 July, with slopes of 1.25 and 1.08, respectively (1.5 and 1.3 after adjusting for the calibration difference). These two days were characterized by the highest mixing ratios of $O_3$, isoprene, and the anthropogenic VOCs ethene and ethyne. This linear fit of all the data is difficult to interpret, however, since the $XO_2$ measurements are 30 minute averages and the $HO_2^*$ measurements are 1-minute averages taken every 30 minutes. A regression of the binned data shown in Fig. 5 gives the relation $[XO_2] = 0.98\,[HO_2^*] + 1.7$ ppt; accounting for the calibration difference gives an adjusted slope of 1.18.

To further investigate the effect of this different averaging on the comparison, on 22 July the IU-LIF-FAGE instrument operated in $HO_2^*$-only mode (i.e., with no time devoted to measuring OH). We compare the resulting 1-minute and 15-minute averaged $HO_2^*$ measurements to the 1.5 minute and 15-minute averaged $XO_2$ measurements (Fig. 7). Between 15:00 and 17:00, the $HO_2^*$ measurements increased from 50 to 70 pptv and decreased back to 50 pptv while the $XO_2$ measurements were relatively invariant at 40 pptv. Ignoring the difference between the average mixing ratios, this difference in the temporal profile of the two instruments' measurements result could only be "real" if there were changes in the peroxy radical relative composition on this two-hour time scale, e.g. a simultaneous increase in $HO_2$ and a decrease in alkyl peroxy radicals, such that $[HO_2^*]$ actually did increase while the mixing ratio of total peroxy radicals was almost constant. Measurements of VOC composition and NOx do not support such a fast change in peroxy radical composition, suggesting that these observations were more likely the result of an instrumental issue. Currently we are unable to identify the exact cause of this observation, but possible explanations are a transient interference in the $HO_2^*$ measurement when sampling ambient air or a change in the sensitivity of the ECHAMP measurements.

The "true" $[XO_2]/[HO_2^*]$ ratio, i.e., the ratio that would be produced by the two instruments' measurements if they were calibrated to the same source and operated exactly as expected without any uncharacterized interferences or losses, depends on the composition of the peroxy radicals. As described in Section 2 (Experimental Methods), for both ECHAMP and LIF-FAGE, the sensitivity of the instrument to individual $RO_2$ compounds depends on the R-group and is characterized by the parameter "$\alpha$", which is the instrument's sensitivity to each $RO_2$ relative to its sensitivity to $HO_2$. For ECHAMP $\alpha$ is determined largely by the fraction of $RO_2$ that is converted to alkyl nitrates ($RONO_2$) and alkyl nitrites ($RONO$) following reaction with NO at atmospheric pressure. For LIF-FAGE, $\alpha$ is mostly determined by how quickly each $RO_2$ is converted sequentially to $HO_2$ and then OH following reaction with NO after the expansion of the sampled gas into the low-pressure region of the instrument (Fuchs et al., 2011;Lew et al., 2018). Air in which $CH_3O_2$, $CH_3C(O)O_2$, and small (<C5) alkyl peroxy radicals have a large contribution to the total peroxy radical concentration would thus produce a relatively high



[XO$_2$]/[HO$_2$*] value, since ECHAMP is sensitive to those peroxy radicals ($\alpha$>0.9) whereas the LIF-FAGE HO$_2$* measurement is not ($\alpha$<0.1). In contrast, air with a relatively high fraction of alkene-derived RO$_2$ (e.g., isoprene peroxy radicals), for which both ECHAMP and LIF-FAGE HO$_2$* $\alpha$ values are near one, would be expected to lead to lower [XO$_2$]/[HO$_2$*] values (i.e., closer to unity).

5       Because the composition of the peroxy radicals during IRRONIC is not exactly known, we examine the predicted speciation generated by zero-dimensional photochemical modeling of the IRRONIC dataset using two versions of the Regional Atmospheric Chemistry Mechanism (RACM2 and RACM2-LIM1) and the Master Chemical Mechanism (MCM 3.2 and 3.3.1). A full comparison of the modeled and measured concentrations is beyond the scope of this paper; we use these model outputs mainly to inform the discussion of the relative speciation of total peroxy radicals and its relation to the expected and measured [XO$_2$]/[HO$_2$*] ratio. A fuller description of the photochemistry at this site, including OH reactivity measurements, will be described in a companion paper (Lew et al, in preparation).

       The accuracy of the model results is, of course, subject to how comprehensive and accurate the supporting measurements and underlying chemical mechanisms are, but nonetheless help to frame the interpretation of the two instruments' measurements. Due to gaps in the NO data because of problems with the Thermo chemiluminescence sensor, there are only three days for which we have model results and measured peroxy radical concentrations by both ECHAMP and LIF-FAGE – on the 16th, 22nd, and 24th of July. The model was run for these three days, and also a diurnal profile for the entire campaign was run using diurnal average concentrations of constrained species. From these model results we calculate the expected values measured by ECHAMP and LIF-FAGE based on each instrument's relevant values for $\alpha$:

20   ECHAMP [XO$_2$]$_{EXPECTED}$ = [HO$_2$] + 0.9([CH$_3$O$_2$]) + 0.92([C$_5$H$_8$(OH)O$_2$]) + 0.9([CH$_3$C(O)O$_2$]) + 0.9(Other)     (4)

LIF-FAGE [HO$_2$*]$_{EXPECTED}$ = [HO$_2$] + 0.05([CH$_3$O$_2$]) + 0.83([C$_5$H$_8$(OH)O$_2$]) + 0.05([CH$_3$C(O)O$_2$]) + 0.7(Other)     (5)

The "Other" category includes all types of peroxy radicals, e.g., from monoterpenes, ethene, etc.

25       The top portion of Fig. 5 shows the average diurnal profile for the modeled and measured [XO$_2$]/[HO$_2$*] ratio using all days when there were both XO$_2$ and HO$_2$* measurements. Between 10:00 and 18:00 the modeled [XO$_2$]/[HO$_2$*] ratio using RACM2 varied between 1.2 and 1.4, whereas the measured ratio varied between 0.9 and 1.4, with a greater amount of variability from hour to hour. Increasing the observed ratio by 20% to account for the calibration comparison (section 3.1) gives an adjusted measured ratio of between 1.1 and 1.7. The highly variable ratios during nighttime mainly reflect the lower signal to noise ratios of both instruments when peroxy radical concentrations were low (less than ~5 ppt).

30       Measured and RACM2 modeled concentrations for 16, 22, and 24 July are shown in Fig. 8. On all three days the relative contributions from the various types of peroxy radicals are comparable. At 15:30 –when concentrations are highest – the modeled peroxy radicals comprised 40% C$_5$H$_8$(OH)O$_2$, 33% HO$_2$, 16% CH$_3$O$_2$, 4% CH$_3$C(O)O$_2$, and 7% "Other", resulting in an expected (modeled) [XO$_2$]/[HO$_2$*] ratio of 1.3. The measured [XO$_2$]/[HO$_2$*] ratio is close to unity on 16 and



22 July, and between 1.2 and 1.5 on 24 July. Increasing these measured ratios by 20% to account for the calibration comparison results produces adjusted measured $[XO_2]/[HO_2*]$ ratios of 1.2 on 16 and 22 July and 1.4 to 1.8 on 24 July.

Measured $[XO_2]$ mixing ratios are ~40% lower than the modeled $[XO_2]$ on 16 and 22 July but within 10% on 24 July. The comparison between measured $[HO_2*]$ and modeled $[HO_2*]$ for these three days exhibits more variability (Fig. 8), but except for a few short time periods modeled $[HO_2*]$ is greater than measured $[HO_2*]$. Choice of chemical mechanism does lead to differences in both the absolute concentrations and relative speciation. Compared to RACM2, RACM2-LIM1 leads to 20% higher peak concentrations though the relative speciation is very similar. Concentrations produced by MCM v3.2 are similarly 20% higher than RACM2, though $CH_3C(O)O_2$ accounts for 9% of the total peroxy radical concentration in the afternoon compared to 4% for RACM2. MCM v3.3.1 leads to the highest overall concentrations (~30% higher than RACM2), a 12% contribution from $CH_3C(O)O_2$, and only a 19% contribution from $C_5H_8(OH)O_2$. Further details can be found in the SI. In summary, the $[XO_2]/[HO_2*]$ ratios modeled by the four chemical mechanisms vary from 1.3 (RACM2 and RACM2-LIM1) to 1.4 (MCM v. 32) to 1.5 (MCM v. 331) between 15:00 and 16:00, whereas the measured ratios varied from 0.8 to 1.3 depending on the day, with an average value of 1.0. After accounting for the 20% calibration difference, the modeled and measured ratios agree to within the experimental and model uncertainties. That both the measured $[XO_2]$ and $[HO_2*]$ concentrations, however, are typically only 50% of the values predicted by MCM v3.3.1 does suggest that the model somehow over predicts peroxy radical concentrations, in agreement with past results from high isoprene environments (Griffith et al., 2013;Whalley et al., 2011;Mao et al., 2012).

Observations of $[XO_2]/[HO_2*]$ ratios less than one were observed during parts of 13, 17, and 18 July and even after increasing by 20% to account for the calibration comparison do not seem reasonable or in some cases even possible. These observations were most likely caused by issues with one or both instruments. Two possible causes that warrant investigation in subsequent field measurements are discussed below:

*1. Error in the ECHAMP calibration, especially for RH values greater than 45%.* Although the calibration comparison presented in section 3.1 show that the ECHAMP and LIF-FAGE instrument's calibrations agreed to within measurement uncertainties, that is not necessarily true for RH values greater than those used during those calibration tests. The highest RH value during the calibration comparisons was 45%, whereas the daytime minimum RH values between 12:00 and 16:00, when measured $[XO_2]$ and $[HO_2*]$ were both highest, were typically between 45% and 65% (Fig 1). Furthermore, we cannot prove that the ECHAMP calibration was invariant from day to day. We include potential sampling losses to be a part of the overall ECHAMP calibration.

*2. Interferences in the LIF-FAGE measurement.* The comparison of high temporal resolution in Fig. 7 revealed differences in the temporal profile of the LIF-FAGE and ECHAMP sensor. If these were caused by an interference in the LIF-FAGE measurement when sampling ambient air, then it would follow that the two instruments would agree when sampling a calibration source but differ when sampling ambient air.

As discussed earlier, the RH-dependence of the sensitivity of chemical amplifiers has recently been questioned (Andrés-Hernández et al., 2010;Sommariva et al., 2011). Had we ignored the RH dependence for ECHAMP's amplification factor and simply used the value under dry conditions, the daytime $XO_2$ values would have been roughly 50% lower than those

presented in this paper, leading to unrealistically low $[XO_2]/[HO_2*]$ ratios of ~0.5.

## 4. Conclusions

The results of this comparison of the IU calibration source and the ambient measurements of peroxy radicals by ECHAMP and LIF-FAGE provide encouraging first results that the newly developed ECHAMP method can be used for ambient measurements of total peroxy radicals. The ECHAMP measurements, based on the acetone photolysis method, and the IU

water vapor photolysis calibration source agreed within 12%, within the experimental uncertainties. The measured mixing ratios of $XO_2$ and $HO_2*$ were lower than the concentrations predicted by the RACM2, RACM2-LIM1, MCM v. 3.2, and MCM v. 3.3.1 chemical mechanisms. The measured $[XO_2]/[HO_2*]$ ratios, however, differed from the ratios predicted by zero-dimensional photochemical modeling by less than the combined measurement and modeling uncertainties.

       An attribute of these comparison exercises is that the two instruments operate on very different measurement

principles and the calibration methods differ greatly. Although the calibration comparison was favorable, due to the time required to conduct successful calibrations with the acetone photolysis method and its overall inconvenience (Wood and Charest, 2014) we have discontinued its use. For subsequent field measurements we have used the water vapor photolysis method and another method based on methyl iodide photolysis (Anderson et al., 2018;Clemitshaw et al., 1997;Liu and Zhang, 2014). All three calibration methods do indicate that a humidity-dependent calibration must be used for both CO-

based and ethane-based chemical amplifiers.

## Data Availability

Data are available upon request from the corresponding author (Ezra.Wood@drexel.edu)

## Author contributions.

EW and PS designed the research project. SK and EW were responsible for the ECHAMP measurements and supporting measurements of NO, $NO_2$, and $O_3$. ML, BB, PR, and PS were responsible for the LIF-FAGE measurements and photochemical modeling. SD, SS, TL, and NL were responsible for the measurements of VOCs. SK and EW conducted the analysis and wrote the paper with feedback from all co-authors.





**Competing Interests**. The authors declare that they have no conflicts of interest.

**Acknowledgements.** This study was supported by the National Science Foundation via grant NSF AGS-1443842 to the
University of Massachusetts, NSF grant AGS-1719918 to Drexel University, and NSF grant AGS-1440834 to Indiana
University. This work was also supported by grants from the Regional Council Nord–Pas-de-Calais through the
MESFOZAT project, as well as the French National Research Agency (ANR–11–LABX–0005–01) and the European
Regional Development Fund (ERDF) through the CaPPA (Chemical and Physical Properties of the Atmosphere) project. We
are grateful to J. Flynn and B. Lefer of the University of Houston for the spectroradiometer data (used for the chemical
modelling).



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





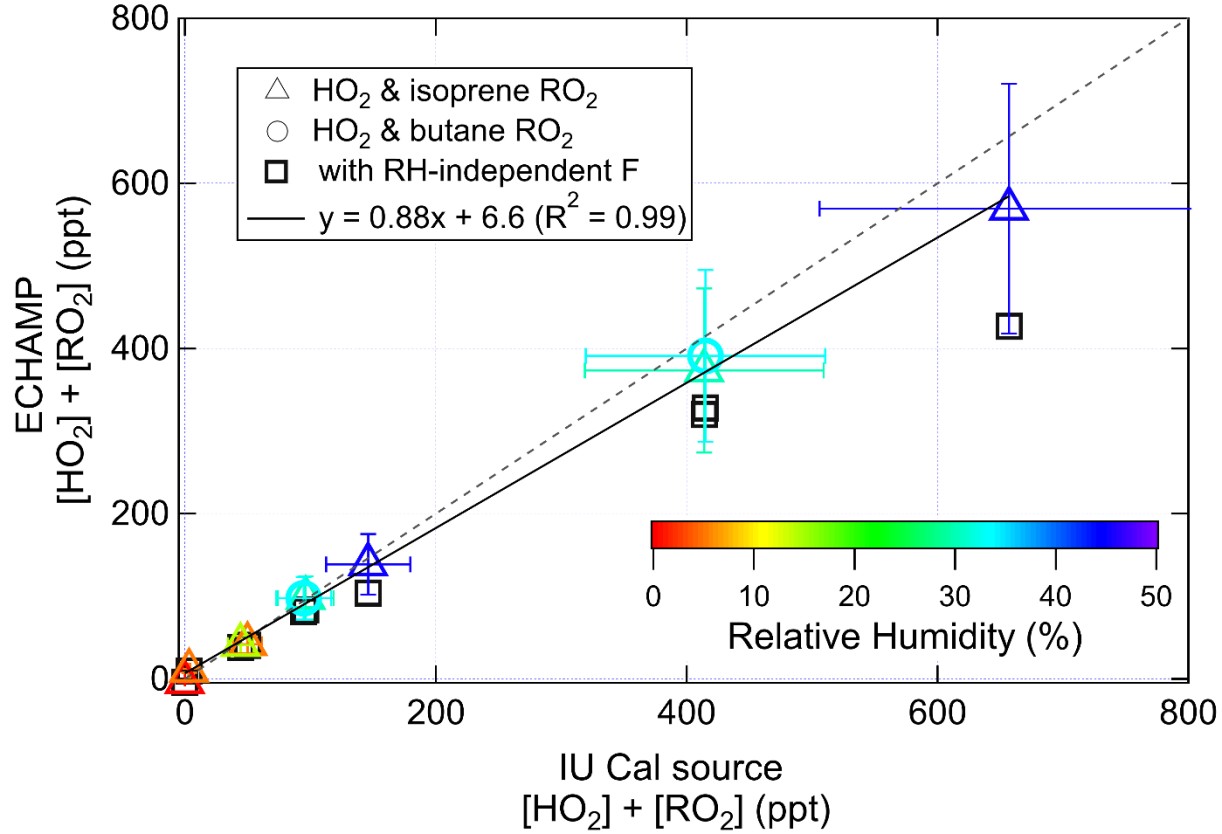

**Figure 2. Results of the calibration comparison in which ECHAMP measured the total peroxy radical concentration in the output of the IU calibration source. The error bars indicate 2σ uncertainties of the ECHAMP measurements and IU calibration source, adjusted for the fact that the IU actinometry was based on the ECHAMP NO₂ calibration. The slope of the dotted line is unity.**



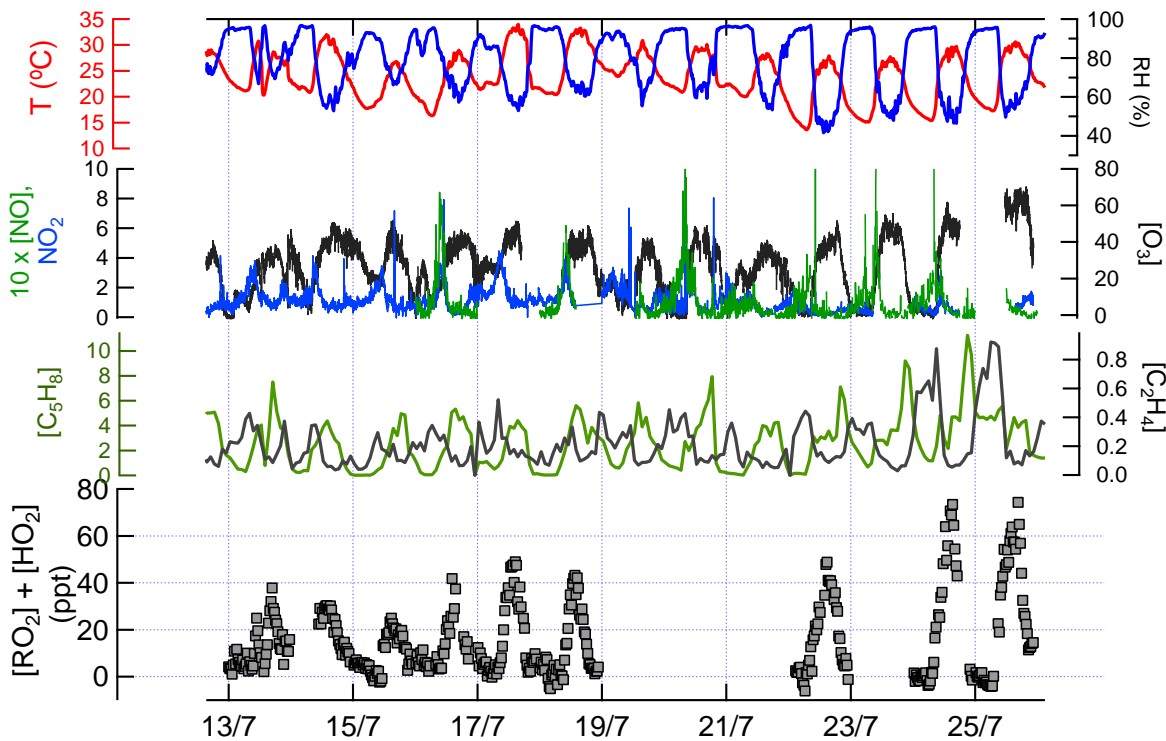

**Figure 3. Time series data of measured chemical and physical parameters during IRRONIC. Except where noted, all measurements are in ppb.**





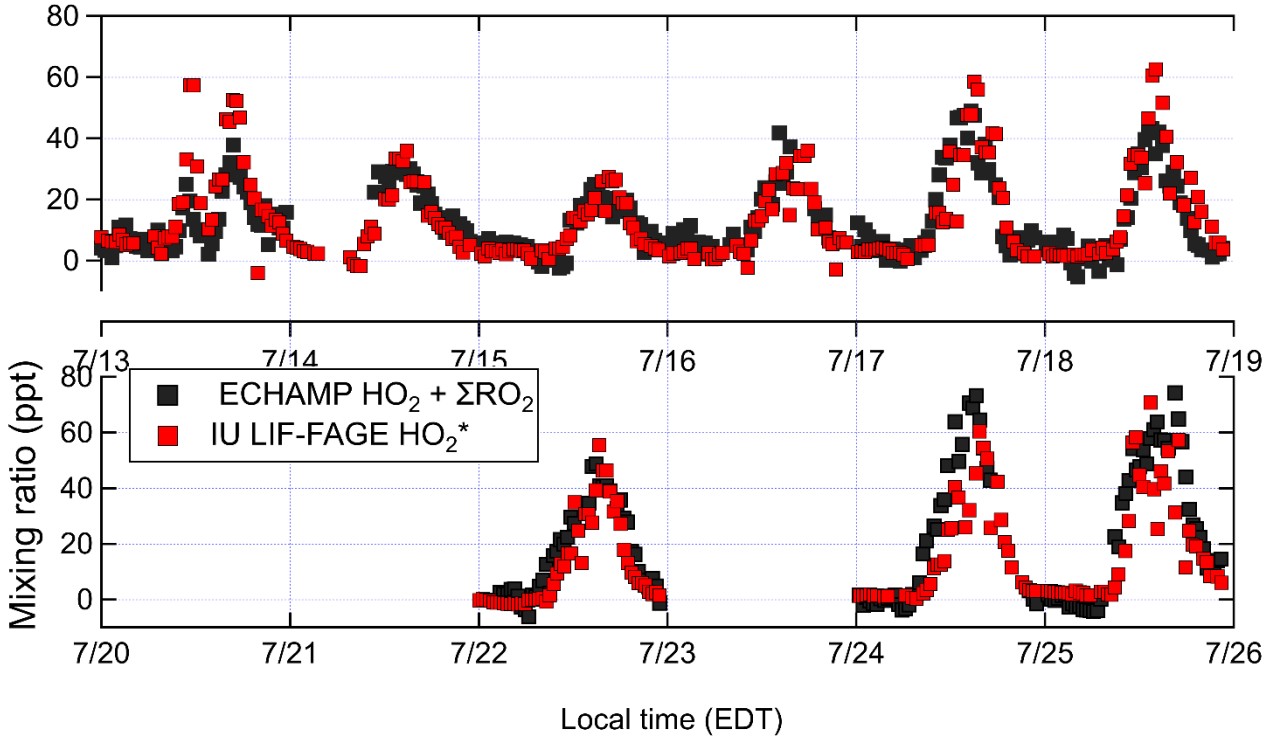

**Figure 4. Concentrations of ambient total peroxy radicals (XO₂) by ECHAMP and HO₂\* by IU-LIF-FAGE. 30-minute averaged measurements are shown for ECHAMP XO₂. For HO₂\*, measurements are 1-minute averages every 30 minutes.**





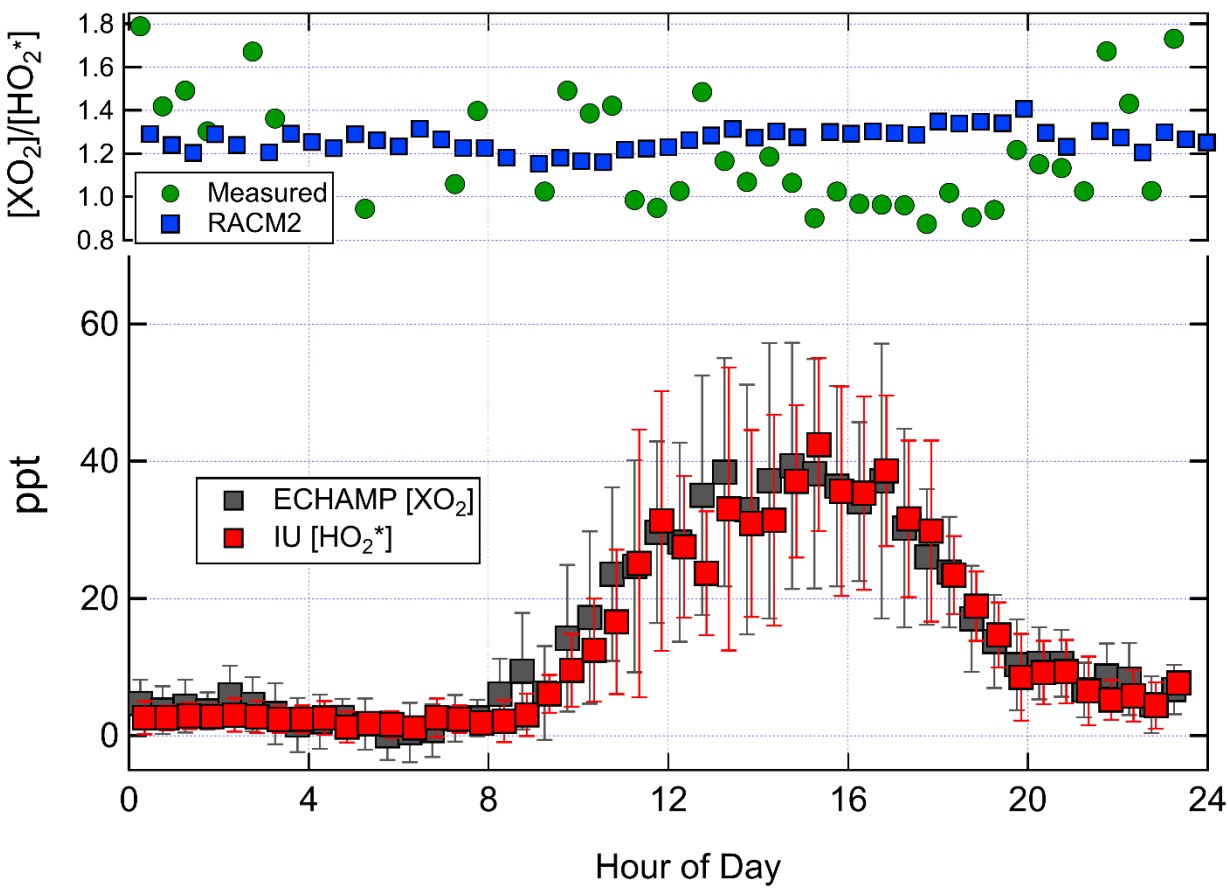

**Figure 5. Lower plot: Mean diurnal profile of ECHAMP XO₂ and IU-LIF-FAGE HO₂\* measurements for the 9 days in which both instruments were operational. The HO₂\* values are displayed with a 6 minute horizontal offset for clarity. The error bars indicate the ± 1σ precision of the measurements in each 30-minute time bin. The upper plot shows the [XO₂]/[HO₂\*] ratio - both measured by the two instruments and modeled using the RACM2 chemical mechanism.**




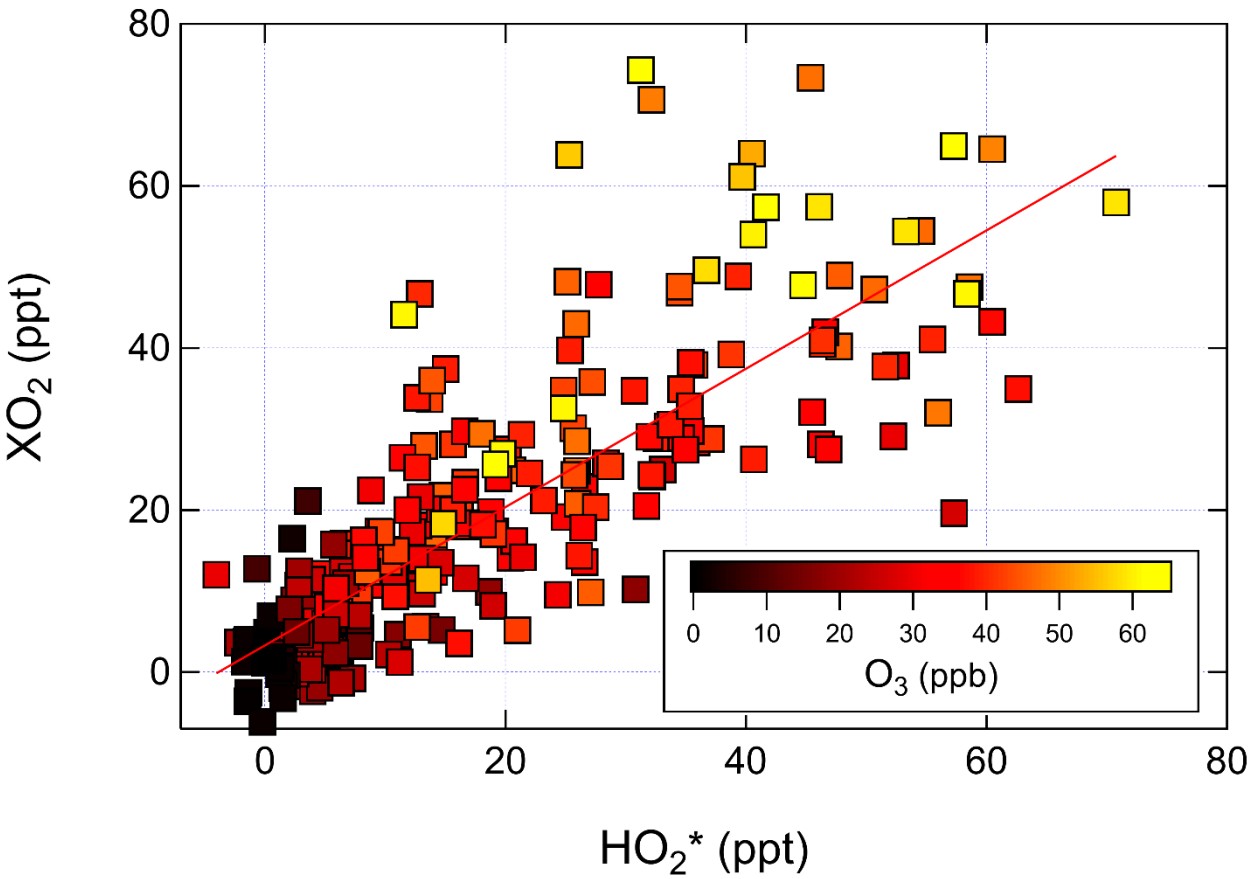

**Figure 6. Correlation of ambient [XO₂] measured by ECHAMP with [HO₂*] measured by IU-LIF-FAGE.**





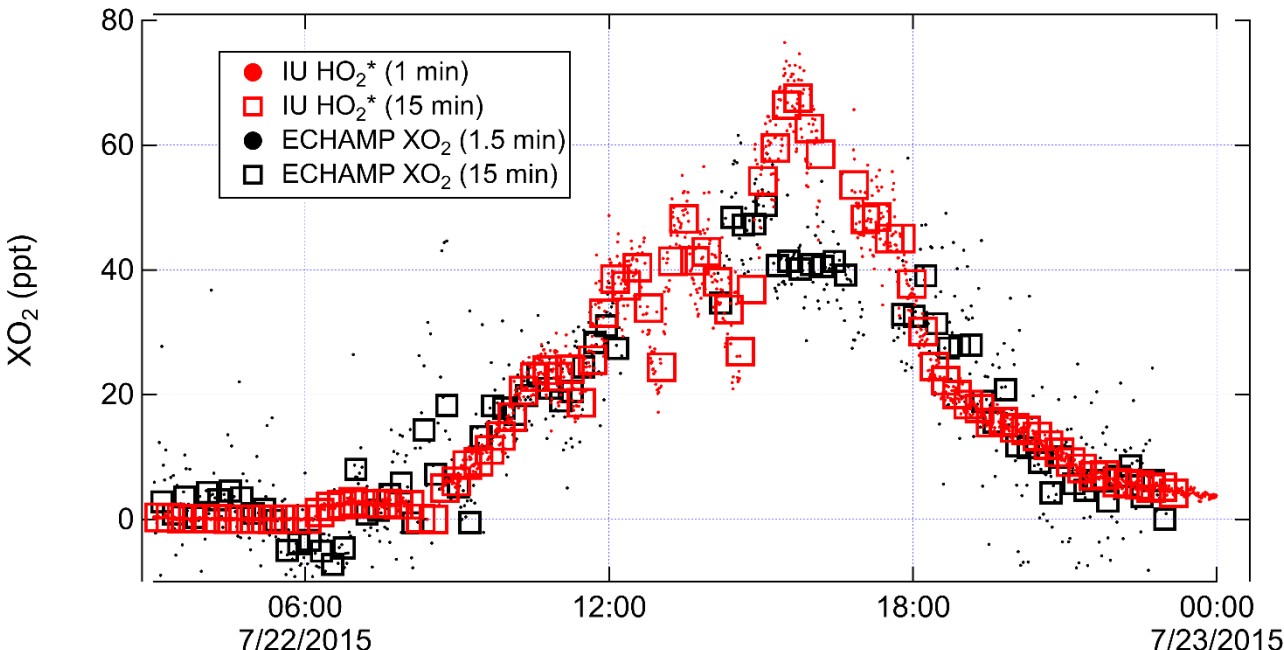

**Figure 7. Time series comparing IU LIF-FAGE HO₂\* and ECHAMP XO₂ measurements from 22 July, 2015 when the IU LIF-FAGE instrument was run in HO₂\*-only mode.**





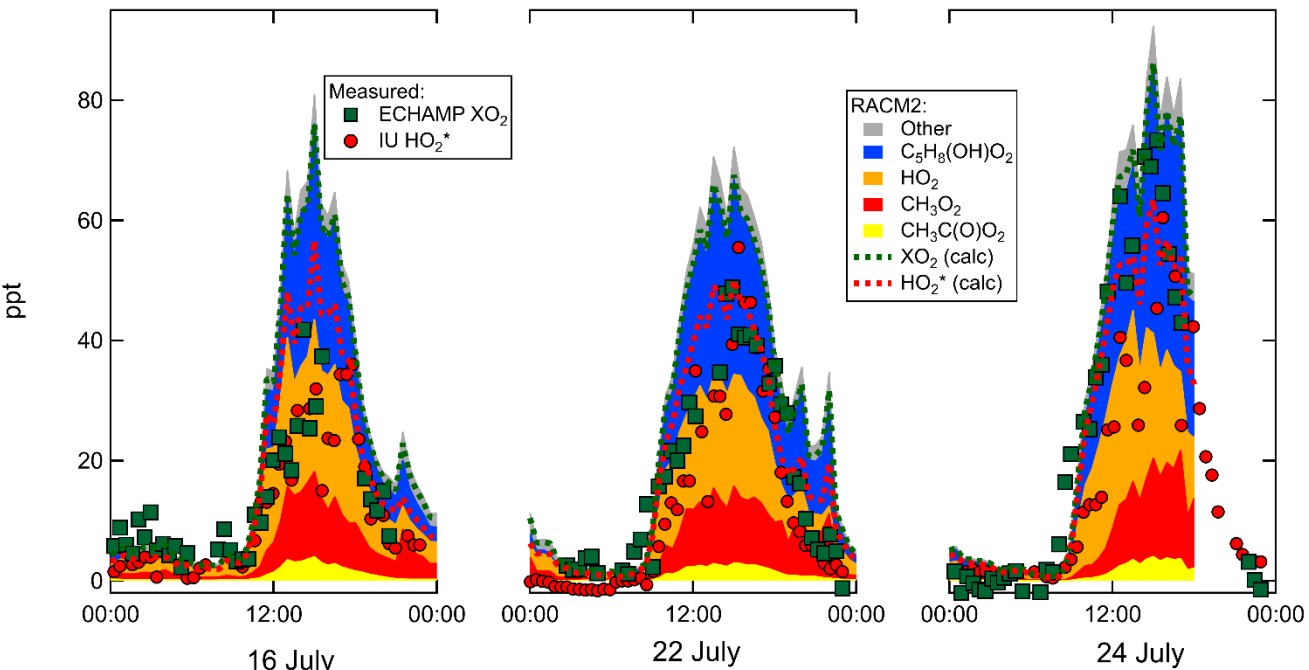

**Figure 8. Peroxy radical mixing ratios measured by ECHAMP and LIF-FAGE and modeled by RACM2.**

