# Peer review of "Peroxy Radical Measurements by Ethane - Nitric Oxide Chemical Amplification and Laser-Induced Fluorescence / Fluorescence Assay by Gas Expansion during the IRRONIC field campaign in a Forest in Indiana"

_Atmospheric Chemistry and Physics, 2018_

## Referee Comment (RC1) · Anonymous Referee #1 · 11 Feb 2019

This manuscript details the results of an intercomparison carried out in the field which compares total peroxy radicals using a chemical amplification system with HO2* (which comprises HO2 and a fraction of RO2 radicals) measured by the FAGE technique. Although we may expect HO2* and total RO2 to be well correlated, the comparison presented here is the detected sum of ambient RO2 by two instruments which do not measure different RO2 radicals with the same efficiency, and so is a tricky undertaking. The authors have employed a variety of models with differing chemical mechanisms to

predict the composition of peroxy radicals present and from there predict the ratio of total RO2 : HO2* for comparison with the observations. I think on the whole, the approach taken to compare these two observations has led to a meaningful comparison and has demonstrated the performance of the new ECHAMP instrument in the field. I recommend publication once the following comments have been addressed:

Abstract: One concerning result is that the XO2:HO2* ratio is periodically less than one, which the authors themselves note is not possible and must indicate a problem with one or both instruments. A ratio of 0.8 is mentioned in the abstract, but no comment on this low ratio is given until the final pages of the manuscript. I suggest the authors are more upfront about this problem and comment on ratios <1 indicating instrumental issues in the abstract and conclusion.

Pg 3, line 18: 'Measurements of OH by laser-induced fluorescence technique can be affected by a sampling related interference which can exceed the actual concentration of OH..'. This interference is very much dependent on the FAGE instrument design. There are several FAGE instruments in operation that do not observe an OH interference and so this statement needs to be qualified to make this clear.

Pg 3, line 25: '$\sim$90%' is slightly misleading. In many of the FAGE instruments tested, $\alpha$ is not as high as 90% for the $\beta$-hydroxy peroxy radicals; $\alpha$ was as low as 17% in the cited 'Whalley et al., 2013' paper.

Pg 7, line 16: please provide the typical Li used for the ambient measurements. SI, section S3: Could the authors comment on whether the loss rate of radicals is solely dependent on residence time? Does the shape of the sampling cross and the PFA tees (the sampled air has to flow around corners) impact the loss rate? This could perhaps be determined if the transmitted radical signal was plotted against residence time in the 4 lengths of tube. An intercept would indicate additional losses in the cross piece.

Pg 7, line 23: The authors discuss the impact of alkyl nitrate and alkyl nitrite formation on the sensitivity of ECHAMP to individual RO2 species, but could the authors also

comment on the expected sensitivity of ECHAMP to RO2 species which are generated from alkene + NO3 reactions, so contain an NO3-adduct? ROxLIF instruments are expected to have a low sensitivity to these types of RO2 (Whalley et al. ACP, 2018). If a similarly low sensitivity for these RO2 is expected in ECHAMP, could the authors discuss how this may influence the measured vs modelled ratio during the night?

Pg 8, line 20: Could the authors make it clear which conversion efficiencies were measured and which have been estimated.

Pg 8, line 22: The authors reference the Fuchs et al., 2011 work on RO2 interferences in FAGE instruments. '$\alpha$' is very much dependent upon the specific FAGE instrument and experimental conditions used, however. Using $\alpha$ determined using another FAGE instrument would likely bias the HO2* model measurement comparison. The authors need to make it clear how $\alpha$ was estimated for the RO2 species not experimentally tested with the IU-FAGE. Specifically, how was $\alpha$ = 0.7 derived in equation 5 on page 14, line 21?

Pg 12, line 12, fig 3: Add the limit of detection of XO2 to the figure. Also make it clear in the figure caption which instrument measured HO2+RO2

Pg 12, section 3.3: The authors acknowledge that comparing the 30 min averaged ECHAMP measurements to a single FAGE measurement made during the 30 minute bin is not ideal. I worry that this approach could introduce bias into the comparison, given that the peroxy radical concentrations will generally be increasing throughout the morning hours and then decreasing during the afternoon and evening. Are the FAGE HO2* measurements made at the midpoint of each 30 minute bin? Does the gradient XO2 vs HO2* vary if the FAGE measurement falls at the start of a 30 minute bin? I think the authors need to explore the robustness of this averaging approach used for the ECHAMP data to satisfy the reader that the two measurements are comparable at the times they are taken.

Page 13, line 6 - 8: the authors report the highest XO2:HO2* ratio on days when

isoprene and ethene concentrations were most elevated. This is unexpected, given the high sensitivity of FAGE to alkene-derived RO2 species. Could the authors comment on this finding?

Page 13, line 10: The data in figure 5 has already been binned and then averaged over 9 days. Does the linear regression on the figure 5 data provide a reduced uncertainty relative to the data presented in figure 6? Errors on the fit should be included. I may have misunderstood, but don't both linear regressions use the same data (just one if further averaged into a diurnal)? Does the change in the regression slope as the data is averaged further suggest that the binning approach is biasing the correlation? Page 13, line 23: Although I appreciate that the authors do not know the reason why the measurements diverge on the 22nd, the possible explanation 'a transient interference in the HO2* measurement when sampling ambient air..' is rather vague. Could the authors elaborate on what they think this transient interference may be or what it may be related to?

Pg 13, line 25 –Pg 14, line 4: I suggest moving this paragraph to the start of section 3.3. It is important that the $\alpha$ of two instruments to different RO2, and how the ratio is expected to change as ambient RO2 types vary, is set out at the beginning of this section. Section 3.3: in general, there is a lot to consider when comparing HO2* and XO2 measured and modelled. The ratio varies with RO2 type present and calibration differences also need to be considered. A table detailing the measured HO2*, XO2 and XO2:HO2* and the 4 modelled HO2*, XO2 and XO2:HO2* on the individual days and campaign average would help to clarify the text.

Figure 4: The caption on the figure is obscuring the top x-axis

Figure 5: There does not seem to be a measured ratio for each 30 min point? Between the hours of 4 – 8, there are only 3 points?

Figure 4 – Figure 8: It is unclear whether the ECHAMP data has been corrected for the calibration comparison or not? This should be clear in each figure caption

[Figure]

---

## Referee Comment (RC2) · Anonymous Referee #2 · 13 Feb 2019

Review of "Peroxy Radical Measurements by Ethane – Nitric Oxide Chemical Amplification and Laser-Induced Fluorescence/Fluorescence Assay by Gas Expansion during the IRRONIC field campaign in a Forest in Indiana" by S. Kundu et al.

The paper describes the measurement of peroxy radicals (HO2, RO2) with two different techniques. The LIF-FAGE technique by Indiana University was originally designed to measure solely HO2 radicals by chemical conversion with NO to OH, which is then

detected by LIF. However, different experimental studies (including a study by authors of this paper) have shown that the technique is also sensitive to specific RO2 radicals with different sensitivities when the instrument is tuned for maximum HO2-to-OH conversion efficiency. The measured quantity is called HO2*. The new ECHAMP technique, a chemical amplifier using ethane instead of CO, is designed to measure the sum of HO2 and all RO2 species. Due to different amplifier chain lengths for different radical species, the resulting quantity XO2 is a proxy for the total peroxy radical concentration. Comparing the measurements by the two techniques sounds like comparing apples with oranges. The present paper demonstrates that such a comparison can be done in a meaningful way, if the instruments are carefully characterized and additional information about the peroxy radical speciation is available (here from box model calculations constrained by measured trace gases). The direct comparison of the conceptually different calibration methods (photolysis of water vapor vs. photolysis of acetone) and the field comparison show that the measurement techniques yield consistent data within the specified experimental uncertainties. These findings suggest that the two described methods can also be used for meaningful tests of atmospheric chemistry models, if the measured peroxy radicals (HO2*, XO2) are appropriately simulated by the model by taking RO2-specific weighting factors of the instruments into account. This requirement should be explicitly stated in the conclusions. Furthermore, recent progress in the measurement of HO2 by LIF-FAGE instruments should be mentioned. It has been shown that the interference by RO2 can be avoided by reducing the concentration of NO that is used for conversion to OH (e.g. Fuchs et al., 2011; Whalley et al., 2013; Feiner et al., 2016; Tan et al., 2017).

Overall, the paper is thoroughly and well written. It is suitable for ACP, but could have been submitted to AMT as well. The authors and editor should consider whether the paper should appear as a "Technical note" in ACP. I recommend publication after the following minor comments have been addressed.

Minor comments

(1) Introduction: as the topic of the paper is an instrumental intercomparison, I suggest to provide a more complete list of previous intercomparisons. For instance, Mount et al. (JGR vol.102, no.D5, p6437, 1997), Zenker et al. (JGR vol 103, no DII, p13,615, 1998), Ren et al. (JGR vol 108, no D19, 4605, 2003), Fuchs et al. (AMT 5, 1611–1626, 2012), Onel et al. (AMT 10, 4877–4894, 2017), Sanchez et al. (Atmos. Env. 174, 227–236, 2018).

(2) In the experimental section, the authors point out that the use of ethane instead of CO offers advantages. Safer operation is obviously a plus. However, I don't understand why the choice of ethane reduces the sensitivity on relative humidity. Is this due to the reduced chain length? Is there evidence for water influence on the OH+CO reaction? To my knowledge, the water effect has been attributed to the reaction HO2+NO (e.g., Mihele et al. 1999, Butkovskaya et al., 2007). Why is the amplification factor lower, if ethane is used? Another advantage of ethane could be mentioned, although it is probably not relevant in a forest environment. Ethane avoids possible interferences from ClOx, which can lead to amplification in CO/NO systems (Perner et al., J. Atmos. Chem. 34, 9, 1999).

(3) Page 9: "For this project, [O3] was instead quantified by the ECHAMP CAPS NO2 sensors after conversion to NO2 by reaction with excess NO". A few details should be explained: is the flow in the calibrator laminar or turbulent? Where is the NO added (upstream, downstream of the calibrator)? Is the NO2 measured after it has been passed through the FAGE cell or is it measured in the air that bypasses the inlet of the FAGE cell? How much NO is added and how large is the resulting NO2 mixing ratio?

(4) Page 9, line 18: is the water vapor correction based on laboratory characterization of the LIF-FAGE instrument, or on theoretical calculations using published data for the OH fluorescence lifetime and cross sections for quenching?

(5) Model constraints: was atmospheric CO measured? Which formaldehyde data were used (GC-FID or DNPH)? I see large gaps in the measured time series of NO

in the first half of the campaign. Was NO (when available) used as a constraint, or was NO calculated by the model using NO2 as a constraint? The box model was constrained with 30 minute average mixing ratios. As peroxy radicals show a strong non-linear dependence on NO, using 30 minute average values as constraint can lead to systematic bias in the model results. I would like to see the model results that are averaged to 30 minutes after the model has been run at the much higher time resolution of the NOx measurements.

(6) Figure 4 - 6: Is it meaningful to adjust the result of the linear regression for the calibration difference (section 3.1)? This would only make sense, if the calibration would be done for the same peroxy radical speciation as encountered during the measurement days in the field.

(7) Figure 1: what is the scale of the map?

(8) Figure 3: what is causing the noise and spikes on the NO data? Is it measurement precision or atmospheric variability from nearby NO sources?

(9) Figure 3 and 4: vertical dotted lines = midnight ?

(10) Figure 5: the shown error bars (1sigma precisions) seem too large compared to the variability of the shown data.

---

## Editor Comment (EC1) · Steven Brown (Editor) · 22 Mar 2019

After the conclusion of the open discussion phase of this paper, I received the following comment from Dr. Maria Dolores Andrés-Hernández. Please address the following comments regarding the representation of the water vapor correction for PERCA instruments in the current literature as part of the revised manuscript.

The manuscript entitled Peroxy Radical Measurements by Ethane - Nitric Oxide Chem-

ical Amplification and Laser-Induced Fluorescence / Fluorescence Assay by Gas Expansion during the IRRONIC field campaign in a Forest in Indiana, by Shuvashish Kundu et al, quotes twice the ACP paper Andrés-Hernández et al., 2011 in a completely wrong context and with an erroneous interpretation of the conclusions of this work.

Kundu et al write in the introduction:

Similarly, XO2 measurements from two CO-based chemical amplifiers during the airborne African Monsoon 5 Multidisciplinary Analysis (AMMA) campaign differed by factors of 2-4 when the usual relative humidity-dependent calibration (Mihele and Hastie, 1998) was used for the chemical amplifier data (Andrés-Hernández et al., 2010).

As a result, the relative humidity dependence of the chemical amplification technique has been questioned (Andrés-Hernández et al., 2010;Sommariva et al., 2011) despite strong experimental evidence (Butkovskaya et al., 2007;Butkovskaya et al., 2005;Butkovskaya et al., 2009;Mihele et al., 1999;Mihele and Hastie, 1998).

And again at the end of the discussion:

As discussed earlier, the RH-dependence of the sensitivity of chemical amplifiers has recently been questioned (AndrésHernández et al., 2010;Sommariva et al., 2011).

This is certainly not true in the case of Andrés-Hernández et al., 2010. In this work it has never been questioned the relative humidity dependency of the amplification factor (chain length: CL) in the chemical amplification. In that context I also recommend the reading and quoting of a previous work of the same group: L. Reichert, M.D Andrés Hernández et al., JGR, 2003, discussing potential mechanisms for the humidity dependency discovered by Mihele and Hastie 1998, without any trace of this effect being questioned by the authors.

I would like to emphasise that in the section 2.1.1. on the Andrés-Hernández et al, 2010 publication is written:

The known dependency of the CL on the relative humidity (RH) of the air sampled (Mihele and Hastie, 1998; Mihele et al., 1999; Reichert et al., 2003) has a negligible effect under the AMMA measurement conditions

A careful reading of the text helps to understand that this statement refers to the particular case of a developed PeRCA instrument for airborne measurements. During the mentioned AMMA campaign two different PeRCA instruments were used:

a) the DUALER from the University of Bremen deployed on the German Falcon, consisting of an inlet kept at constant pressure lower to the ambient. As explained in the text, this minimise humidity in the reactor and consequently its effect on the radical conversion.

b) The PERCA 4 of the University of Leicester deployed on the British FAAM-BAe-146. This instrument neither controlled the pressure during the flight or considered the RH in its calibration and had non solved instrumental issues during the intercomparison exercise. Overall the PERCA 4 measured unrealistic radical values by a factor of 4. This did not question the effect of water vapour on the chain length but the performance of this particular instrument. Therefore only the measurements of the RO2* DUALER were used for the further comparison with the HO2 measured by the FAGE instrument on board of FAAM.

Concerning the quotation of Sommariva et al. 2011, there is in this publication the same kind of wrong interpretation of the results of Andrés Hernández et al., 2010. I came unfortunately across this paper after its publication. Though I contacted the first author Sommariva to clarify his wrong interpretation, the statements in that publication were never corrected.

The analysis of some of the instrumental results presented in the manuscript of Kundu et al., does seem not to fully take into consideration the long term experience and knowledge of the PERCA and LIF radical communities. As mentioned above it is too late now to get involved in the interactive discussion of the manuscript.

But it should not be too late to prevent the use of a wrong interpretation of previous scientific work for supporting questionable results/ instrumental characterisations. It cannot be given the wrong message to the community that the dependency of the chemical amplification on the water vapour is in any form questioned.
* * *

---

## Author Comment (AC1) · 5 May 2019

We thank Dr. Andrés-Hernández for the comments and providing the opportunity to further discuss this important issue. Our responses are interspersed below (in non-italic font).

*The manuscript entitled Peroxy Radical Measurements by Ethane - Nitric Oxide Chemical Amplification and Laser-Induced Fluorescence / Fluorescence Assay by Gas Expansion during the IRRONIC field campaign in a Forest in Indiana, by Shuvashish Kundu et al, quotes twice the ACP paper Andrés-Hernández et al., 2011 in a completely wrong context and with an erroneous interpretation of the conclusions of this work.*

*Kundu et al write in the introduction:*

*"Similarly, XO2 measurements from two CO-based chemical amplifiers during the airborne African Monsoon 5 Multidisciplinary Analysis (AMMA) campaign differed by factors of 2-4 when the usual relative humidity-dependent calibration (Mihele and Hastie, 1998) was used for the chemical amplifier data (Andrés-Hernández et al., 2010).*

*As a result, the relative humidity dependence of the chemical amplification technique has been questioned (Andrés-Hernández et al., 2010;Sommariva et al., 2011) despite strong experimental evidence (Butkovskaya et al., 2007;Butkovskaya et al., 2005;Butkovskaya et al., 2009;Mihele et al., 1999;Mihele and Hastie, 1998)."*

*And again at the end of the discussion:*

*"As discussed earlier, the RH-dependence of the sensitivity of chemical amplifiers has recently been questioned (Andrés Hernández et al., 2010;Sommariva et al., 2011)."*

*This is certainly not true in the case of Andrés-Hernández et al., 2010. In this work it has never been questioned the relative humidity dependency of the amplification factor (chain length: CL) in the chemical amplification. In that context I also recommend the reading and quoting of a previous work of the same group: L. Reichert, M.D Andrés Hernández et al., JGR, 2003, discussing potential mechanisms for the humidity dependency discovered by Mihele and Hastie 1998, without any trace of this effect being questioned by the authors.*

*I would like to emphasise that in the section 2.1.1. on the Andrés-Hernández et al, 2010 publication is written:*
*The known dependency of the CL on the relative humidity (RH) of the air sampled (Mihele and Hastie, 1998; Mihele et al., 1999; Reichert et al., 2003) has a negligible effect under the AMMA measurement conditions.*

*A careful reading of the text helps to understand that this statement refers to the particular case of a developed PeRCA instrument for airborne measurements. During the mentioned AMMA campaign two different PeRCA instruments were used:*

*a) the DUALER from the University of Bremen deployed on the German Falcon, consisting of an inlet kept at constant pressure lower to the ambient. As explained in the text, this minimise humidity in the reactor and consequently its effect on the radical conversion.*

*b) The PERCA 4 of the University of Leicester deployed on the British FAAM-BAe-146. This instrument neither controlled the pressure during the flight or considered the RH in its calibration and had non solved instrumental issues during the intercomparison exercise. Overall the PERCA 4 measured unrealistic radical values by a factor of 4. This did not question the effect of water vapour on the chain length but the performance of this particular instrument. Therefore only the measurements of the RO2\* DUALER were used for the further comparison with the HO2 measured by the FAGE instrument on board of FAAM.*

*Concerning the quotation of Sommariva et al. 2011, there is in this publication the same kind of wrong interpretation of the results of Andrés Hernández et al., 2010. I came unfortunately across this paper after its publication. Though I contacted the first author Sommariva to clarify his wrong interpretation, the statements in that publication were never corrected.*

We have carefully re-read Andrés-Hernandez et al. (2010) to investigate the apparent confusion over this issue. As described in the short comment, that paper describes the operation of two chemical amplifier (PERCA) instruments: the U. of Bremen DUALER instrument and the U. of Leicester PERCA4 instrument. The RH-dependence of the CL is described for both instruments:

1. For the U. Bremen DUALER instrument, in section 2.1.1 of that paper, the authors describe the U. of Bremen DUALER instrument and state "The known dependency of the CL on the relative humidity (RH) of the air sampled (Mihele and Hastie, 1998; Mihele et al., 1999; Reichert et al., 2003) has a negligible effect under the AMMA measurement conditions." This conclusion is based on the fact that because the instrument reactor's pressure was lower and temperature higher than ambient conditions, the RH is kept below 15%, a range in which the variability of the chain length is apparently smaller than the error in the CL determination: "The RH in the reactors remained below 15%. At 300 mbar this effect in the CL is expected to be within the error of the CL determination". This last sentence is likely what initially confused us, since we would expect the effect to be experimentally quantified rather than use an "expected" effect. Furthermore, at atmospheric pressure the CL of the Bremen chemical amplifier is almost 40% lower at 15% RH compared to 0% RH (Reichert et al., 2003)! Even if the uncertainty in the CL determination is comparable or larger, this would probably be irrelevant since those uncertainties would cancel as they are correlated assuming they stem from the usual spectroscopic uncertainties (e.g., the absorption cross sections for $O_2$ and $H_2O$ used for the water vapor photolysis calibration method). For example, if the CL at 0% RH is $100 \pm 30\%$ and the CL at 15% RH is $80 \pm 30\%$ (hypothetical numbers), that would still be a 20% decrease in sensitivity! Kartal et al. (2010), which describes the Bremen DUALER instrument in detail, does not show the RH dependence of the DUALER at 300 mbar.

We now recognize that the Bremen team did not question the RH dependence – rather they stated that it had a negligible effect as operated during those conditions. We remain confused how the change in CL between 0 and 15% RH can be considered negligible.

2. For the Leicester PERCA instrument, calibrations were only performed at ground level and not at altitude, which the authors recognize was not preferable. To correct for the effect of RH and the decreased pressure and temperature at aircraft altitudes, computational results were used.

When this computationally-based P, T, and RH-dependent chain length was applied to the data, however, very high concentrations results –more than four times higher than the DUALER peroxy radical measurements, higher than the LIF-FAGE $HO_2$ measurements, and higher than the modeled concentrations. Moreover, a trend in the modeled to measured peroxy radical concentration was observed when using the RH-corrected data, but not when using the uncorrected data. The authors conclude the following:

> *"To apply the humidity correction to this dataset may therefore be inappropriate and the intercomparison was made using the uncorrected data"*

The authors hypothesize that the reason for this was condensation of water on their inlet, which would greatly decrease the RH inside their reaction chamber.

The 2$^{nd}$ paper we quoted – Sommariva et al (2011) – clearly rejected the RH-correction to their instrument:

> *"…the determination of CL during the experiment with the PAN source was carried out in ambient (humid) air. The agreement between the CL determined in ambient air with PAN and the CL determined in zero air with $CH_3I$ demonstrates that the data collected during the TexAQS 2006 cruise did not require a correction for relative humidity. This was also the case for a similar PERCA instrument during another field campaign under similarly high ambient humidity and temperature conditions (Andres-Hernandez et al., 2010)."*

Based on these two papers in aggregate, we reasonably concluded that the RH effect had been questioned. This is certainly the case for Sommariva et al. (2011). For Andres-Hernandez et al. (2010), we now see that we were incorrect to state that that paper questioned the RH dependence. We had interpreted their results - that the RH effect was expected to be negligible when their past results showed almost a 40% difference between 0 and 15% RH – to mean that the RH effect was in some way questioned. In the revision of our paper we will clarify that Sommariva et al. (2011) questioned the RH effect but Andres-Hernandez (2010) did. We will mention that the RH effect is addressed in a variety of ways, including that described by Andres-Hernandez et al. (2010) and Kartal et al. (2010), both of which stated that it is negligible for their airborne measurements.

*The analysis of some of the instrumental results presented in the manuscript of Kundu et al., does seem not to fully take into consideration the long term experience and knowledge of the PERCA and LIF radical communities. As mentioned above it is too late now to get involved in the interactive discussion of the manuscript. But it should not be too late to prevent the use of a wrong interpretation of previous scientific work for supporting questionable results/ instrumental characterisations.*

These last few comments we find puzzling. We recognize the concerns about the RH dependence as discussed above. Without explicit explanation of which aspects of our analysis comparing the ECHAMP and LIF results from July 2015 the commenter finds questionable, we are unable to respond to the criticism. The new ECHAMP instrument very much owes its existence to the prior work conducted by both the PERCA and LIF communities. ECHAMP's predecessor instrument

(Wood and Charest, 2014) was a chemical amplifier that used the same CO and NO-based chemistry as all prior PERCA instruments (e.g., Cantrell and Stedman, 1982;Clemitshaw et al., 1997;Kartal et al., 2010), used the same anti-synchronized dual-channel design pioneered by the Leicester PERCA group (Green et al., 2006), and addressed the RH dependence first identified by Mihele and Hastie (1998) and studied in detail by others (Butkovskaya et al., 2007;Butkovskaya et al., 2005;Butkovskaya et al., 2009;Reichert et al., 2003). Its current version (Anderson et al., 2019) uses the $O_3$ chemical actinometry variation of the water vapor photolysis calibration method introduced by a chemical amplifier group (Schultz et al., 1995) and used by several PERCA and LIF groups (Bloss et al., 2004;Dusanter et al., 2008;Kartal et al., 2010). It also now uses a methyl iodide-based calibration source, similar to that described by other PERCA groups (Clemitshaw et al., 1997;Liu and Zhang, 2014). Furthermore, the Indiana University LIF-FAGE group plays a major role in this manuscript!

*It cannot be given the wrong message to the community that the dependency of the chemical amplification on the water vapour is in any form questioned.*

In our revision we will clarify that the RH dependence has not been questioned by the Bremen group, and only attribute that to Sommariva et al. (2011) who certainly did question the RH dependence, evident from the text quoted earlier. Sommariva et al. (2011) also appeared to ignore the large body of work on the homogeneous reaction between NO and the $HO_2$-$H_2O$ adduct (Butkovskaya et al., 2007;Butkovskaya et al., 2005;Butkovskaya et al., 2009;Mihele and Hastie, 1998;Reichert et al., 2003) in their comment below as part of the open discussion of their paper:

*"The mechanism of the humidity interference remains unclear, although it appears to be heterogeneous (i.e., wall loss) in origin."*

We hope that our revised paper, including this helpful discussion, clarifies that the RH-dependence must be addressed by all chemical amplifier instruments.

References
Anderson, D. C., Pavelec, J., Daube, C., Herndon, S. C., Knighton, W. B., Lerner, B. M., Roscioli, J. R., Yacovitch, T. I., and Wood, E. C.: Characterization of Ozone Production in San Antonio, Texas, Using Measurements of Total Peroxy Radicals, Atmos. Chem. Phys., 19, 2845-2860, 10.5194/acp-19-2845-2019, 2019.

Andrés-Hernández, M. D., Stone, D., Brookes, D. M., Commane, R., Reeves, C. E., Huntrieser, H., Heard, D. E., Monks, P. S., Burrows, J. P., Schlager, H., Kartal, D., Evans, M. J., Floquet, C. F. A., Ingham, T., Methven, J., and Parker, A. E.: Peroxy radical partitioning during the AMMA radical intercomparison exercise, Atmos. Chem. Phys., 10, 10621-10638, 10.5194/acp-10-10621-2010, 2010.

Bloss, W. J., Lee, J. D., Bloss, C., Heard, D. E., Pilling, M. J., Wirtz, K., Martin-Reviejo, M., and Siese, M.: Validation of the calibration of a laser-induced fluorescence instrument for the

measurement of OH radicals in the atmosphere, Atmospheric Chemistry and Physics, 4, 571-583, 2004.

Butkovskaya, N., Kukui, A., Pouvesle, N., and Le Bras, G.: Formation of nitric acid in the gas-phase $HO_2$ + NO reaction: Effects of temperature and water vapor, J. Phys. Chem. A, 109, 6509-6520, 2005.

Butkovskaya, N., Kukui, A., and Le Bras, G.: $HNO_3$ Forming Channel of the $HO_2$ + NO Reaction as a Function of Pressure and Temperature in the Ranges of 72-600 Torr and 223-323 K, J. Phys. Chem. A, 111, 9047-9053, 2007.

Butkovskaya, N., Rayez, M.-T., Rayez, J.-C., Kukui, A., and Le Bras, G.: Water vapor effect on the $HNO_3$ yield in the $HO_2$ + NO reaction: experimental and theoretical evidence, J. Phys. Chem. A, 113, 11327-11342, 2009.

Cantrell, C., and Stedman, D.: A possible technique for the measurement of atmospheric peroxy radicals, Geophys. Res. Let., 9, 846-849, 1982.

Clemitshaw, K. C., Carpenter, L. J., Penkett, S. A., and Jenkin, M. E.: A calibrated peroxy radical chemical amplifier for ground-based tropospheric measurements, J. Geophys. Res., 102, 25405, 10.1029/97jd01902, 1997.

Dusanter, S., Vimal, D., and Stevens, P. S.: Technical note: Measuring tropospheric OH and $HO_2$ by laser-induced fluorescence at low pressure. A comparison of calibration techniques, Atmos. Chem. Phys., 8, 321-340, 10.5194/acp-8-321-2008, 2008.

Green, T. J., Reeves, C. E., Fleming, Z. L., Brough, N., Rickard, A. R., Bandy, B. J., Monks, P. S., and Penkett, S. A.: An improved dual channel PERCA instrument for atmospheric measurements of peroxy radicals, Journal of Environmental Monitoring, 8, 530, 10.1039/b514630e, 2006.

Kartal, D., Andrés-Hernández, M. D., Reichert, L., Schlager, H., and Burrows, J. P.: Technical Note: Characterisation of a DUALER instrument for the airborne measurement of peroxy radicals during AMMA 2006, Atmos. Chem. Phys, 10, 3047-3062, 2010.
Liu, Y., and Zhang, J.: Atmospheric Peroxy Radical Measurements using Dual-Channel Chemical Amplification Ravity Ringdown Spectroscopy, Analytical Chemistry, 86, 5391-5398, 2014.

Mihele, C. M., and Hastie, D. R.: The sensitivity of the radical amplifier to ambient water vapour, Geophys. Res. Let., 25, 1911-1913, 10.1029/98gl01432, 1998.

Reichert, L., Hernández, A., Stöbener, D., Burkert, J., and Burrows, J.: Investigation of the effect of water complexes in the determination of peroxy radical ambient concentrations: Implications for the atmosphere, J. of Geophys. Res., 108, 2003.

Schultz, M., Heitlinger, M., Mihelcic, D., and Volz-Thomas, A.: Calibration source for peroxy radicals with built-in actinometry using $H_2O$ and $O_2$ photolysis at 185 nm, J. Geophys. Res., 100, 18811 - 18816, 1995.

Sommariva, R., Brown, S. S., Roberts, J. M., Brookes, D. M., Parker, A. E., Monks, P. S., Bates, T. S., Bon, D., de Gouw, J. A., Frost, G. J., Gilman, J. B., Goldan, P. D., Herndon, S. C., Kuster, W. C., Lerner, B. M., Osthoff, H. D., Tucker, S. C., Warneke, C., Williams, E. J., and Zahniser, M. S.: Ozone production in remote oceanic and industrial areas derived from ship based measurements of peroxy radicals during TexAQS 2006, Atmos. Chem. Phys., 11, 2471-2485, 2011.

Wood, E. C., and Charest, J.: Chemical Amplification – Cavity Attenuated Phase Shift Spectrometer Measurements of Peroxy Radicals, Anal. Chem., 86, 10266-10273, 2014.

---

## Author Comment (AC2) · 21 May 2019

We thank both reviewers for their comments. Before addressing their concerns, we note a few other issues:

1. An error was discovered in how the 0-D photochemical modeling was conducted. The revised model results are used in the revision. The conclusions of the paper remain unchanged. The main difference between the original modeling and the revised modeling is that the relative speciation of peroxy radicals predicted by the four chemical mechanisms (RACM2, RACM2-LIM1, MCM 3.2, and MCM 3.3.1) is now very similar as described in the following two paragraphs from section 3.3 of the manuscript:

"Measured and MCM 3.2 modeled concentrations for 16, 22, and 24 July are shown in Fig. 8. On all three days the relative contributions from the various types of peroxy radicals are comparable. At 15:30 –when concentrations are highest – the modeled peroxy radicals comprised 30% $C_5H_8(OH)O_2$, 35% $HO_2$, 26% $CH_3O_2$ and 7% $CH_3C(O)O_2$. The four chemical mechanisms vary little in the predicted relative speciation (SI). The $[XO_2]/[HO_2*]$ ratio modeled by MCM 3.2 between 15:00 and 16:00 is 1.4 for 16 and 22 July and 1.45 on 24 July. The measured $[XO_2]/[HO_2*]$ ratio is close to unity on 16 and 22 July, and between 1.2 and 1.5 on 24 July. Increasing these measured ratios by 20% to account for the calibration comparison produces adjusted measured $[XO_2]/[HO_2*]$ ratios of 1.2 on 16 and 22 July and 1.4 to 1.8 on 24 July. After accounting for the 20% calibration difference, the modeled and measured ratios agree to within the experimental and model uncertainties.

Measured $[XO_2]$ mixing ratios are 20 to 30% lower than the MCM 3.2 $[XO_2]$ on 16 and 22 July but agree more closely on 24 July (measured/modeled ratio varies from 0.8 to 1.15). The comparison between measured $[HO_2*]$ and modeled $[HO_2*]$ for these three days exhibits more variability (Fig. 8). Although all four chemical mechanisms predict a very similar relative speciation, there are variations in the absolute peroxy radical concentrations predicted. MCM 3.3.1 concentrations are very similar to those from MCM 3.2, but RACM2 and RACM2-LIM1 predict 26% and 42% higher peak concentrations, respectively. Further details can be found in the SI."

2. We have revised the following section in order to address the open comment from Dr. Andres-Hernandez:

"Similarly, $XO_2$ measurements from two CO-based chemical amplifiers during the airborne African Monsoon Multidisciplinary Analysis (AMMA) campaign differed by factors of 2-4 when the usual relative humidity-dependent calibration (Mihele and Hastie, 1998) was used for the chemical amplifier data, though the performance of one of the instruments was not assessed with in-flight calibrations (Andrés-Hernández et al., 2010). The relative humidity dependence of the chemical amplification technique is addressed in a variety of ways. Most research groups characterize their instrument's amplification factor (chain length) as a function of relative humidity (RH) which they then apply to their measurements based on the ambient RH. In some cases, because the RH in the amplification chamber can be lower than ambient because of reduced pressure and higher temperatures, the variability in RH can be considered negligible compared to other experimental uncertainties (Andrés-Hernández et al., 2010;Kartal et al., 2010). In one case the need to apply an RH-dependent calibration was disputed (Sommariva et al., 2011) despite strong experimental evidence (Butkovskaya et al., 2007;Butkovskaya et al., 2005;Butkovskaya et al., 2009;Mihele et al., 1999;Mihele and Hastie, 1998;Reichert et al., 2003)."

3. We have changed the method by which the linear fit was determined for figure 6 – only data between 09:00 and 22:00 are now used due to the low signal-to-noise of the nighttime measurements. This is further described on page 5 of this document.

Our responses to the comments from reviewer #1 are below:

*This manuscript details the results of an intercomparison carried out in the field which compares total peroxy radicals using a chemical amplification system with HO2\* (which comprises HO2 and a fraction of RO2 radicals) measured by the FAGE technique. Although we may expect HO2\* and total RO2 to be well correlated, the comparison presented here is the detected sum of ambient RO2 by two instruments which do not measure different RO2 radicals with the same efficiency, and so is a tricky undertaking. The authors have employed a variety of models with differing chemical mechanisms to predict the composition of peroxy radicals present and from there predict the ratio of total RO2 : HO2\* for comparison with the observations. I think on the whole, the approach taken to compare these two observations has led to a meaningful comparison and has demonstrated the performance of the new ECHAMP instrument in the field. I recommend publication once the following comments have been addressed:*

*Abstract: One concerning result is that the XO2:HO2\* ratio is periodically less than one, which the authors themselves note is not possible and must indicate a problem with one or both instruments. A ratio of 0.8 is mentioned in the abstract, but no comment on this low ratio is given until the final pages of the manuscript. I suggest the authors are more upfront about this problem and comment on ratios <1 indicating instrumental issues in the abstract and conclusion.*

We have added the following sentence to the abstract:

"Time periods in which the ambient ratio was less than one are definitely caused by measurement errors (including calibration differences) as such ratios are not physically possible."

and to the abstract:

"The measured $[XO_2]/[HO_2\*]$ ratios usually differed from the ratios predicted by zero-dimensional photochemical modeling by less than the combined measurement and modeling uncertainties, though the lowest ratios observed (0.8) are not physically meaningful and therefore must be due to measurement errors."

*Pg 3, line 18: 'Measurements of OH by laser-induced fluorescence technique can be affected by a sampling related interference which can exceed the actual concentration of OH..'. This interference is very much dependent on the FAGE instrument design. There are several FAGE instruments in operation that do not observe an OH interference and so this statement needs to be qualified to make this clear.*

We have edited that sentence as follows:

"Measurements of OH by the laser-induced fluorescence technique can be affected by a sampling-related interference which can exceed the actual concentration of OH (Mao et al., 2012), though the magnitude of this interference and even its presence varies greatly depending on instrument design."

*Pg 3, line 25: '~90%' is slightly misleading. In many of the FAGE instruments tested, α is not as high as 90% for the β-hydroxy peroxy radicals; α was as low as 17% in the cited 'Whalley et al., 2013' paper.*

We have edited that section as follows:

"The sensitivity of the LIF-FAGE technique to each type of organic peroxy radical varies with the amount of NO added for the conversion and is instrument-dependent but in general is highest (up to ~90%) for β–hydroxy peroxy radicals derived from alkenes and lowest for those derived from small alkanes (Fuchs et al., 2011;Lew et al., 2018;Whalley et al., 2013). This $RO_2$ interference can be greatly reduced by use of lower NO concentrations or reaction times, yielding conversion efficiencies for isoprene-$RO_2$ under 20% (Feiner et al., 2016;Fuchs et al., 2011;Tan et al., 2017;Whalley et al., 2013)."

*Pg 7, line 16: please provide the typical Li used for the ambient measurements.*

The previous sentence has been edited to clarify that equation 2 is not used to calculate ambient measurements:

"Including a sampling loss term, the sensitivity "α" of ECHAMP to individual organic peroxy radicals relative to that of $HO_2$ can be estimated using Equation 2:"

Later in that paragraph and in the SI, the sampling losses are described.

*SI, section S3: Could the authors comment on whether the loss rate of radicals is solely dependent on residence time? Does the shape of the sampling cross and the PFA tees (the sampled air has to flow around corners) impact the loss rate? This could perhaps be determined if the transmitted radical signal was plotted against residence time in the 4 lengths of tube. An intercept would indicate additional losses in the cross piece.*

Section S3 describes two types of radical loss tests: 1. measuring the transmission of $HO_2$ through four lengths of tubing, and 2. measuring the $HO_2$ signal when sampling through the sampling cross compared to sampling directly at the reaction chamber. As stated in the SI,

"Similarly, the second method – comparing the ECHAMP signal when sampling a radical source through the sampling cross or directly into one of the reaction chambers – indicated overall losses of less than 4% for an $HO_2$ source."

Further details of the sampling losses (including loss rates onto several types of material) are the subject of a separate manuscript currently under preparation.

*Pg 7, line 23: The authors discuss the impact of alkyl nitrate and alkyl nitrite formation on the sensitivity of ECHAMP to individual RO2 species, but could the authors also comment on the expected sensitivity of ECHAMP to RO2 species which are generated from alkene + NO3 reactions, so contain an NO3-adduct? ROxLIF instruments are expected to have a low sensitivity to these types of RO2 (Whalley et al. ACP, 2018). If a similarly low sensitivity for these RO2 is expected in ECHAMP, could the authors discuss how this may influence the measured vs modelled ratio during the night?*

We have not yet conducted experiments in the lab to determine the sensitivity of ECHAMP to RO2 produced from $NO_3$ reactions, though we do expect that when mixed with NO and ethane these peroxy radicals will lead to increases in $NO_2$. We have added the following sentence regarding the sensitivity of ECHAMP to these radicals at the end of section 2.2:

"We estimate an elevated uncertainty of ~50% for the measurements at night as we have not investigated the sensitivity of ECHAMP to peroxy radical produced by ozonolysis and $NO_3$ reactions."

We are reluctant to comment on the measured vs. modelled ratio during the night for two reasons: 1. The measurements (of both peroxy radicals and the crucial compound NO) had much lower signal-to-noise ratios and higher uncertainties at night, and 2. The measurement height was only 3 meters which complicates the interpretation of the data since the air was usually stagnant.

*Pg 8, line 20: Could the authors make it clear which conversion efficiencies were measured and which have been estimated.*

The sentence has been re-worded to clarify:

"The conversion efficiencies for other major $RO_2$ radicals are estimated as 5% for $CH_3O_2$ and…"

*Pg 8, line 22: The authors reference the Fuchs et al., 2011 work on RO2 interferences in FAGE instruments. 'α' is very much dependent upon the specific FAGE instrument and experimental conditions used, however. Using α determined using another FAGE instrument would likely bias the HO2\* model measurement comparison. The authors need to make it clear how α was estimated for the RO2 species not experimentally tested with the IU-FAGE. Specifically, how was α = 0.7 derived in equation 5 on page 14, line 21?*

The sentence on pg 5 has been edited as follows:

"The conversion efficiencies for other major $RO_2$ are estimated as 5% for $CH_3O_2$ and the acetyl peroxy radical ($CH_3C(O)O_2$), 8% for ethyl peroxy radical ($C_2H_5O_2$), and 31-55% for $RO_2$ compounds from the OH oxidation of high-molecular-weight hydrocarbons based on comparisons to several other interference tests (Fuchs et al., 2011;Griffith et al., 2016;Lew et al., 2018)."

The value of 0.7 for the "other" category was chosen since light alkanes, which have low values of α, comprise a minor component of the OH reactivity and most other $RO_2$ compounds have α values near 0.7.

We have added the following sentences:

"The α values for ECHAMP are based on the calculated yields of alkyl nitrates and alkyl nitrites as described in section 2.2. For LIF-FAGE, the α value for $C_5H_8(OH)O_2$ was measured and α for $CH_3O_2$ and $CH_3C(O)O_2$ are based on measured yields to several similar instruments all of which have measured values less than 5%. An α of 0.7 is assume for the "other" category since most alkenes have α values between 0.5 and 0.9, and small alkanes, which have lower values, account for a small portion of the OH reactivity (Lew et al., in preparation)."

*Pg 12, line 12, fig 3: Add the limit of detection of XO2 to the figure. Also make it clear in the figure caption which instrument measured HO2+RO₂*

The limit of detection (LOD) depends on the relative humidity and the variability in the ambient ozone concentration as described in Wood et al. 2017. To exactly determine the LOD at any given time requires operating both reaction chambers in background mode, precluding simultaneous knowledge of the exact LOD and the ambient concentrations. Rather than add an estimated limit of detection to the figure, we have added the following text to the caption:

"The sum of $[RO_2]$ and $[HO_2]$ was measured by the ECHAMP instrument, with a detection limit typically between 1 and 2 ppt (signal-to-noise ratio of two)."

*Pg 12, section 3.3: The authors acknowledge that comparing the 30 min averaged ECHAMP measurements to a single FAGE measurement made during the 30 minute bin is not ideal. I worry that this approach could introduce bias into the comparison, given that the peroxy radical concentrations will generally be increasing throughout the morning hours and then decreasing during the afternoon and evening. Are the FAGE HO2\* measurements made at the midpoint of each 30 minute bin? Does the gradient XO2 vs HO2\* vary if the FAGE measurement falls at the start of a 30 minute bin? I think the authors need to explore the robustness of this averaging approach used for the ECHAMP data to satisfy the reader that the two measurements are comparable at the times they are taken.*

We have majorly revised the paragraph below. We have also changed the averaging time used for the linear regression:

"A bi-variate linear regression of the $XO_2$ and $HO_2^*$ measurements between 09:00 and 22:00 yields the relationship $[XO_2] = (1.08 \pm 0.05)$ $[HO_2^*] - (1.4 \pm 0.3)$ ppt (Fig 6.). The regression is restricted to this window of time because of the degraded precision of the ECHAMP measurements at night due to the higher relative humidity. The $[XO_2]/[HO_2^*]$ slopes were highest on the last two days of measurements – 24 and 25 July, with slopes of 1.25 and 1.08, respectively, or 1.5 and 1.3 after adjusting for the calibration difference. These two days were characterized by the highest mixing ratios of peroxy radicals, $O_3$, isoprene, and the anthropogenic VOCs ethene and ethyne. The lowest $[XO_2]/[HO_2^*]$ ratios were observed on 13 July during which a passing thunderstorm led to low concentrations during mid-day with higher values before and after the storm. The higher $[XO_2]/[HO_2^*]$ ratios observed later in the field campaign may simply be the result of a change in sensitivity in one of the instruments. These linear are difficult to interpret, however, since the $XO_2$ measurements are 30 minute averages and the $HO_2^*$ measurements are 1-minute averages taken every 30 minutes. A regression of the binned data shown in Fig. 5 gives the relation $[XO_2] = 1.0 \pm 0.14$ $[HO_2^*] + (1.5 \pm 1.6)$ ppt; accounting for the calibration difference gives an adjusted slope of 1.2. The $[XO_2]/[HO_2^*]$ ratio using the binned data was highest between 9:45 and 10:45 (Fig. 5), but was between 0.9 and 1.1 between 14:45 and 19:15. This overall temporal trend is apparent in several days (Fig. 4). Applying a 30-min offset to the $XO_2$ data largely removes this trend and leads to fewer time periods when $[XO_2]/[HO_2^*]$ was less than 1.0, but such an offset does not agree with the synchronized time-base of both measurements. The two instruments' different averaging times and precision levels preclude further assessment and conclusions regarding possible time offsets."

*Page 13, line 6 - 8: the authors report the highest XO2:HO2\* ratio on days when isoprene and ethene concentrations were most elevated. This is unexpected, given the*

*high sensitivity of FAGE to alkene-derived RO2 species. Could the authors comment on this finding?*

We have edited that sentence:

"These two days were characterized by the highest mixing ratios of $O_3$, isoprene, and the anthropogenic VOCs ethene and ethyne. The high $[XO_2]/[HO_2*]$ ratios observed those days may simply be the result of a change in sensitivity in one of the instruments."

*Page 13, line 10: The data in figure 5 has already been binned and then averaged over 9 days. Does the linear regression on the figure 5 data provide a reduced uncertainty relative to the data presented in figure 6? Errors on the fit should be included. I may have misunderstood, but don't both linear regressions use the same data (just one if further averaged into a diurnal)? Does the change in the regression slope as the data is averaged further suggest that the binning approach is biasing the correlation?*

The binning is useful because there are occasional gaps in the time series (e.g., the morning of 14 July). Without the binning, the morning data is slightly "underrepresented" because of that gap. We have changed the caption as follows (including the fit errors):

"Figure 6. Correlation of ambient $[XO_2]$ measured by ECHAMP with $[HO_2*]$ measured by IU-LIF-FAGE. The linear fit is for data between 09:00 and 22:00, indicated by the points with green circles. The equation of the fit is $[XO_2] = (1.08 \pm 0.05) [HO_2^*] - (1.4 \pm 0.3)$ ppt."

*Page 13, line 23: Although I appreciate that the authors do not know the reason why the measurements diverge on the 22nd, the possible explanation 'a transient interference in the HO2* measurement when sampling ambient air..' is rather vague. Could the authors elaborate on what they think this transient interference may be or what it may be related to?*

We agree that the explanation of a "transient interference" is vague, but feel that any possible reason offered at this point would be too speculative. We note that since $HO_2*$ is measured as OH after conversion by reaction with NO, any interference in the OH measurement would affect the $HO_2*$ measurements as well.

*Pg 13, line 25 –Pg 14, line 4: I suggest moving this paragraph to the start of section 3.3. It is important that the α of two instruments to different RO2, and how the ratio is expected to change as ambient RO2 types vary, is set out at the beginning of this section.*

We agree and have made that paragraph the 2nd paragraph of section 3.3 in the revision.

*Section 3.3: in general, there is a lot to consider when comparing HO2* and XO2 measured and modelled. The ratio varies with RO2 type present and calibration differences also need to be considered. A table detailing the measured HO2*, XO2 and XO2:HO2* and the 4 modelled HO2*, XO2 and XO2:HO2* on the individual days and campaign average would help to clarify the text.*

We hope that the majorly revised paragraph quoted earlier (starting with "A bi-variate linear regression…") has clarified these issues. Furthermore, the results from the 4 models are shown in the SI.

*Figure 4: The caption on the figure is obscuring the top x-axis*
This has been fixed in the revision.

*Figure 5: There does not seem to be a measured ratio for each 30 min point? Between the hours of 4 – 8, there are only 3 points?*

Because of the low signal-to-noise ratios for the nighttime measurements (especially by ECHAMP), the ratio of the measured $XO_2/[HO_2^*]$ varies greatly at night, from 0.3 to 2.1, and so some of those points were off-scale (the graph's axis was from 0.8 to 1.8). For the revision, we only show the ratio for time periods between 08:00 and 22:00, with the following revised caption:

"…The upper plot shows the $[XO_2]/[HO_2^*]$ ratio - both measured by the two instruments and modeled using the MCM 3.2 chemical mechanism. The measured ratio is only shown for time periods between 08:00 and 22:00 due to the poor signal-to-noise ratios for the night-time measurements."

*Figure 4 – Figure 8: It is unclear whether the ECHAMP data has been corrected for the calibration comparison or not? This should be clear in each figure caption*

We have added the following sentence to section 3.3 to clarify:

"No adjustments have been made to either of the datasets in Fig. 4 (or any other figures) to account for the calibration difference."

Reviewer #2's comments:

*The paper describes the measurement of peroxy radicals (HO2, RO2) with two different techniques. The LIF-FAGE technique by Indiana University was originally designed to measure solely HO2 radicals by chemical conversion with NO to OH, which is then detected by LIF. However, different experimental studies (including a study by authors of this paper) have shown that the technique is also sensitive to specific RO2 radicals with different sensitivities when the instrument is tuned for maximum HO2-to-OH conversion efficiency. The measured quantity is called HO2\*. The new ECHAMP technique, a chemical amplifier using ethane instead of CO, is designed to measure the sum of HO2 and all RO2 species. Due to different amplifier chain lengths for different radical species, the resulting quantity XO2 is a proxy for the total peroxy radical concentration. Comparing the measurements by the two techniques sounds like comparing apples with oranges. The present paper demonstrates that such a comparison can be done in a meaningful way, if the instruments are carefully characterized and additional information about the peroxy radical speciation is available (here from box model calculations constrained by measured trace gases). The direct comparison of the conceptually different calibration methods (photolysis of water vapor vs. photolysis of acetone) and the field comparison show that the measurement techniques yield consistent data within the specified experimental uncertainties. These findings suggest that the two*

*described methods can also be used for meaningful tests of atmospheric chemistry models, if the measured peroxy radicals (HO2\*, XO2) are appropriately simulated by the model by taking RO2-specific weighting factors of the instruments into account. This requirement should be explicitly stated in the conclusions.*

*Furthermore, recent progress in the measurement of HO2 by LIF-FAGE instruments should be mentioned. It has been shown that the interference by RO2 can be avoided by reducing the concentration of NO that is used for conversion to OH (e.g. Fuchs et al., 2011; Whalley et al., 2013; Feiner et al., 2016; Tan et al., 2017).*

This section has been re-written as follows:

*"*The sensitivity of the LIF-FAGE technique to each type of organic peroxy radical varies with the amount of NO added for the conversion and is instrument-dependent but in general is highest (up to ~90%) for β–hydroxy peroxy radicals derived from alkenes and lowest for those derived from small alkanes (Fuchs et al., 2011;Lew et al., 2018;Whalley et al., 2013). This $RO_2$ interference can be greatly reduced by use of lower NO concentrations or reaction times, yielding conversion efficiencies for isoprene-$RO_2$ lower than 20% (Feiner et al., 2016;Fuchs et al., 2011;Tan et al., 2017;Whalley et al., 2013).*"*

*Overall, the paper is thoroughly and well written. It is suitable for ACP, but could have been submitted to AMT as well. The authors and editor should consider whether the paper should appear as a "Technical note" in ACP. I recommend publication after the following minor comments have been addressed.*

We agree that the paper could have been suitable for AMT as well. We chose to submit to ACP because we think that the comparison of the measured concentrations with those by the models provided information beyond that of an instrument assessment and provided information on our community's understanding of HOx chemistry in low-NOx, high biogenic VOC environments, which has historically been problematic.

*(1) Introduction: as the topic of the paper is an instrumental intercomparison, I suggest to provide a more complete list of previous intercomparisons. For instance, Mount et al. (JGR vol.102, no.D5, p6437, 1997), Zenker et al. (JGR vol 103, no Dll, p13,615, 1998), Ren et al. (JGR vol 108, no D19, 4605, 2003), Fuchs et al. (AMT 5, 1611–1626, 2012), Onel et al. (AMT 10, 4877–4894, 2017), Sanchez et al. (Atmos. Env. 174, 227–236, 2018).*

We have added the suggested references.

*(2) In the experimental section, the authors point out that the use of ethane instead of CO offers advantages. Safer operation is obviously a plus. However, I don't understand why the choice of ethane reduces the sensitivity on relative humidity. Is this due to the reduced chain length? Is there evidence for water influence on the OH+CO reaction? To my knowledge, the water effect has been attributed to the reaction HO2+NO (e.g., Mihele et al. 1999, Butkovskaya et al., 2007). Why is the amplification factor lower, if ethane is used? Another advantage of ethane could be mentioned, although it is*

*probably not relevant in a forest environment. Ethane avoids possible interferences
from ClOx, which can lead to amplification in CO/NO systems (Perner et al., J. Atmos.
Chem. 34, 9, 1999).*

We have added the following text to briefly clarify the important issue of RH-dependence:

"The cause of the RH-dependence of the CO-based amplification chemistry is the RH-dependence of the main radical termination step: the reaction of $HO_2$ with NO to form $HNO_3$ (Butkovskaya et al., 2007;Butkovskaya et al., 2005;Butkovskaya et al., 2009;Mihele et al., 1999;Reichert et al., 2003), with a smaller contribution from the RH-dependent wall losses of $HO_2$. These two RH-dependent radical termination steps affect the ethane-based amplification chemistry as well, but the most important terminations steps are from the formation of ethyl nitrite and ethyl nitrate – neither of which depends on relative humidity."

*(3) Page 9: "For this project, [O3] was instead quantified by the ECHAMP CAPS NO2
sensors after conversion to NO2 by reaction with excess NO". A few details should be
explained: is the flow in the calibrator laminar or turbulent? Where is the NO added
(upstream, downstream of the calibrator)? Is the NO2 measured after it has been
passed through the FAGE cell or is it measured in the air that bypasses the inlet of the
FAGE cell? How much NO is added and how large is the resulting NO2 mixing ratio?*

We have edited the following section in order to provide more information on this quantification:

"For this project, $[O_3]$ was instead quantified by the ECHAMP CAPS $NO_2$ sensors after conversion to $NO_2$ by reaction with excess NO. This was accomplished by having the IU calibration source overflow the ECHAMP inlet. ECHAMP was operated without the ethane flowing, so that each reaction channel sampled 1 LPM of air from the cal source into which 80 sccm of 21 ppm NO was added. This resulted in a diluted concentration of 1.7 ppm NO, which is high enough to react with 99% of the $O_3$ formed during the transit from the inlet to the CAPS detectors. This produces a very precise measurement of the sum of $[O_3]$ and $[NO_2]$ ($1\sigma$ precision of 22 ppt for 10 second averages). The accuracy of this ozone determination is thus ultimately traceable to the CAPS $NO_2$ calibration (see SI). Typical $[O_3]$ values measured were between 0.4 and 2.0 ppb.**"**

*(4) Page 9, line 18: is the water vapor correction based on laboratory characterization
of the LIF-FAGE instrument, or on theoretical calculations using published data for the
OH fluorescence lifetime and cross sections for quenching?*

It is based on laboratory characterizations. The new sentence:

"The sensitivity of the instrument is corrected for fluorescence quenching by water vapor as per laboratory characterization"

*(5) Model constraints: was atmospheric CO measured? Which formaldehyde data
were used (GC-FID or DNPH)?*

CO was not measured but was estimated based on published emission ratios of CO with benzene. Formaldehyde was only measured using the DNPH cartridges.

The relevant sentences were edited as follows:

"…cartridges to measure carbonyls, including formaldehyde (which was not measured by the GC-FID system), acetaldehyde and…"

and

"Measured VOC concentrations (every 90 min) were interpolated on to this 30 min time resolution. Carbon monoxide was not measured but instead estimated based on emission ratios of CO with benzene (Warneke et al., 2007)."

*I see large gaps in the measured time series of NO in the first half of the campaign. Was NO (when available) used as a constraint, or was NO calculated by the model using NO2 as a constraint?*

The comparison to the models is heavily focused on the three days when there measurements of NO, $XO_2$ (by ECHAMP) and $HO_2$* (by LIF-FAGE) as described in the text:

"Due to gaps in the NO data because of problems with the Thermo chemiluminescence sensor, there are only three days for which we have model results and measured peroxy radical concentrations by both ECHAMP and LIF-FAGE – on the 16[th], 22[nd], and 24[th] of July. The model was run for these three days, and also a diurnal profile for the entire campaign was run using diurnal average concentrations of constrained species."

*The box model was constrained with 30 minute average mixing ratios. As peroxy radicals show a strong non-linear dependence on NO, using 30 minute average values as constraint can lead to systematic bias in the model results. I would like to see the model results that are averaged to 30 minutes after the model has been run at the much higher time resolution of the NOx measurements.*

The time resolution of the model is limited by the 90-minute frequency of the VOC measurements which we have interpolated to values every 30-minutes. Thus we are unable to run the model at higher time resolution.

*(6) Figure 4 - 6: Is it meaningful to adjust the result of the linear regression for the calibration difference (section 3.1)? This would only make sense, if the calibration would be done for the same peroxy radical speciation as encountered during the measurement days in the field.*

We have intentionally included in the text both the "raw" regression/ratio results and those corrected for the calibration difference. Since both ECHAMP and LIF-FAGE are both sensitive (high α) to $HO_2$ and isoprene $RO_2$, we do think that "correcting" the comparisons for the 20% calibration difference helps to frame the discussion of the differences between the two measurements.

*(7) Figure 1: what is the scale of the map?*
The caption has been updated to address this:

"The arrow represents a distance of_1 km."

*(8) Figure 3: what is causing the noise and spikes on the NO data? Is it measurement precision or atmospheric variability from nearby NO sources?*

The data shown in figure 3 are the 5-minute averaged NO concentrations which have a 1σ precision of approximately 100 ppt. The "spikes" in the figure are actually of 15 to 60 minute duration, and thus are from atmospheric variability (mostly during the early morning).

*(9) Figure 3 and 4: vertical dotted lines = midnight ?*

Yes. We have updated the figure captions to clarify:

"The vertical grid lines indicate midnight for odd-numbered days, in local time."

*(10) Figure 5: the shown error bars (1sigma precisions) seem too large compared to the variability of the shown data*

In the caption for figure 5 we had erroneously described the error bars as indicative of the 1σ precision of the measurements when they actually just describe the distribution of the measured concentrations. We have changed that sentence in the caption to the following:

[revised manuscript text omitted]

---

## Author Response (AR1)

**Author response**

In addition to the changes listed in the point-by-point response, the following changes were made:

1. An error was discovered in how the 0-D photochemical modeling was conducted. The revised model results are used in the revision. The conclusions of the paper remain unchanged. The main difference between the original modeling and the revised modeling is that the relative speciation of peroxy radicals predicted by the four chemical mechanisms (RACM2, RACM2-LIM1, MCM 3.2, and MCM 3.3.1) is now very similar as described in the following two paragraphs from section 3.3 of the manuscript:

"Measured and MCM 3.2 modeled concentrations for 16, 22, and 24 July are shown in Fig. 8. On all three days the relative contributions from the various types of peroxy radicals are comparable. At 15:30 – when concentrations are highest – the modeled peroxy radicals comprised 30%  $C_5H_8(OH)O_2$ , 35% HO2, 26%  $CH_3O_2$  and 7%  $CH_3C(O)O_2$ . The four chemical mechanisms vary little in the predicted relative speciation (SI). The  $[XO_2]/[HO_2*]$  ratio modeled by MCM 3.2 between 15:00 and 16:00 is 1.4 for 16 and 22 July and 1.45 on 24 July. The measured  $[XO_2]/[HO_2*]$  ratio is close to unity on 16 and 22 July, and between 1.2 and 1.5 on 24 July. Increasing these measured ratios by 20% to account for the calibration comparison produces adjusted measured  $[XO_2]/[HO_2*]$  ratios of 1.2 on 16 and 22 July and 1.4 to 1.8 on 24 July. After accounting for the 20% calibration difference, the modeled and measured ratios agree to within the experimental and model uncertainties.

Measured [XO2] mixing ratios are 20 to 30% lower than the MCM 3.2 [XO2] on 16 and 22 July but agree more closely on 24 July (measured/modeled ratio varies from 0.8 to 1.15). The comparison between measured [HO2\*] and modeled [HO2\*] for these three days exhibits more variability (Fig. 8). Although all four chemical mechanisms predict a very similar relative speciation, there are variations in the absolute peroxy radical concentrations predicted. MCM 3.3.1 concentrations are very similar to those from MCM 3.2, but RACM2 and RACM2-LIM1 predict 26% and 42% higher peak concentrations, respectively. Further details can be found in the SI."

2. We have revised the following section in order to address the open comment from Dr. Andres-Hernandez:

"Similarly, XO2 measurements from two CO-based chemical amplifiers during the airborne African Monsoon Multidisciplinary Analysis (AMMA) campaign differed by factors of 2-4 when the usual relative humidity-dependent calibration (Mihele and Hastie, 1998) was used for the chemical amplifier data, though the performance of one of the instruments was not assessed with in-flight calibrations (Andrés-Hernández et al., 2010). The relative humidity dependence of the chemical amplification technique is addressed in a variety of ways. Most research groups characterize their instrument's amplification factor (chain length) as a function of relative humidity (RH) which they then apply to their measurements based on the ambient RH. In some cases, because the RH in the amplification chamber can be lower than ambient because of reduced pressure and higher temperatures, the variability in RH can be considered negligible compared to other experimental uncertainties (Andrés-Hernández et al., 2010;Kartal et al., 2010). In one case the need to apply an RH-dependent calibration was disputed (Sommariva et al., 2011) despite strong experimental evidence (Butkovskaya et al., 2007;Butkovskaya et al., 2005;Butkovskaya et al., 2009;Mihele et al., 1999;Mihele and Hastie, 1998;Reichert et al., 2003)." 3. We have changed the method by which the linear fit was determined for figure 6 -only data between 09:00 and 22:00 are now used due to the low signal-to-noise of the nighttime measurements. This is further described on page 5 of this document.

Our responses to the comments from reviewer #1 are below:

This manuscript details the results of an intercomparison carried out in the field which compares total peroxy radicals using a chemical amplification system with HO2\* (which comprises HO2 and a fraction of RO2 radicals) measured by the FAGE technique. Although we may expect HO2\* and total RO2 to be well correlated, the comparison presented here is the detected sum of ambient RO2 by two instruments which do not measure different RO2 radicals with the same efficiency, and so is a tricky undertaking. The authors have employed a variety of models with differing chemical mechanisms to predict the composition of peroxy radicals present and from there predict the ratio of total RO2 : HO2\* for comparison with the observations. I think on the whole, the approach taken to compare these two observations has led to a meaningful comparison and has demonstrated the performance of the new ECHAMP instrument in the field. I recommend publication once the following comments have been addressed:

Abstract: One concerning result is that the XO2:HO2\* ratio is periodically less than one, which the authors themselves note is not possible and must indicate a problem with one or both instruments. A ratio of 0.8 is mentioned in the abstract, but no comment on this low ratio is given until the final pages of the manuscript. I suggest the authors are more upfront about this problem and comment on ratios <1 indicating instrumental issues in the abstract and conclusion.

We have added the following sentence to the abstract:

"Time periods in which the ambient ratio was less than one are definitely caused by measurement errors (including calibration differences) as such ratios are not physically possible."

and to the abstract:

"The measured [XO2]/[HO2\*] ratios usually differed from the ratios predicted by zero-dimensional photochemical modeling by less than the combined measurement and modeling uncertainties, though the lowest ratios observed (0.8) are not physically meaningful and therefore must be due to measurement errors."

*Pg 3, line 18: 'Measurements of OH by laser-induced fluorescence technique can be affected by a sampling related interference which can exceed the actual concentration of OH..'. This interference is very much dependent on the FAGE instrument design. There are several FAGE instruments in operation that do not observe an OH interference and so this statement needs to be qualified to make this clear.*

We have edited that sentence as follows:

"Measurements of OH by the laser-induced fluorescence technique can be affected by a sampling-related interference which can exceed the actual concentration of OH (Mao et al., 2012), though the magnitude of this interference and even its presence varies greatly depending on instrument design."

Pg 3, line 25: '~90%' is slightly misleading. In many of the FAGE instruments tested,  $\alpha$  is not as high as 90% for the  $\beta$ -hydroxy peroxy radicals;  $\alpha$  was as low as 17% in the cited 'Whalley et al., 2013' paper.

We have edited that section as follows:

"The sensitivity of the LIF-FAGE technique to each type of organic peroxy radical varies with the amount of NO added for the conversion and is instrument-dependent but in general is highest (up to ~90%) for  $\beta$ – hydroxy peroxy radicals derived from alkenes and lowest for those derived from small alkanes (Fuchs et al., 2011;Lew et al., 2018;Whalley et al., 2013). This RO2 interference can be greatly reduced by use of lower NO concentrations or reaction times, yielding conversion efficiencies for isoprene-RO2 under 20% (Feiner et al., 2016;Fuchs et al., 2011;Tan et al., 2017;Whalley et al., 2013)."

**Pg* 7, *line* 16: *please provide the typical Li used for the ambient measurements.**

The previous sentence has been edited to clarify that equation 2 is not used to calculate ambient measurements:

"Including a sampling loss term, the sensitivity " $\alpha$ " of ECHAMP to individual organic peroxy radicals relative to that of HO2 can be estimated using Equation 2:"

Later in that paragraph and in the SI, the sampling losses are described.

SI, section S3: Could the authors comment on whether the loss rate of radicals is solely dependent on residence time? Does the shape of the sampling cross and the PFA tees (the sampled air has to flow around corners) impact the loss rate? This could perhaps be determined if the transmitted radical signal was plotted against residence time in the 4 lengths of tube. An intercept would indicate additional losses in the cross piece.

Section S3 describes two types of radical loss tests: 1. measuring the transmission of  $HO_2$  through four lengths of tubing, and 2. measuring the  $HO_2$  signal when sampling through the sampling cross compared to sampling directly at the reaction chamber. As stated in the SI,

"Similarly, the second method – comparing the ECHAMP signal when sampling a radical source through the sampling cross or directly into one of the reaction chambers – indicated overall losses of less than 4% for an HO2 source."

Further details of the sampling losses (including loss rates onto several types of material) are the subject of a separate manuscript currently under preparation.

*Pg* 7, line 23: The authors discuss the impact of alkyl nitrate and alkyl nitrite formation on the sensitivity of ECHAMP to individual RO2 species, but could the authors also comment on the expected sensitivity of ECHAMP to RO2 species which are generated from alkene + NO3 reactions, so contain an NO3-adduct? ROxLIF instruments are expected to have a low sensitivity to these types of RO2 (Whalley et al. ACP, 2018). If a similarly low sensitivity for these RO2 is expected in ECHAMP, could the authors discuss how this may influence the measured vs modelled ratio during the night?

We have not yet conducted experiments in the lab to determine the sensitivity of ECHAMP to RO2 produced from  $NO_3$  reactions, though we do expect that when mixed with NO and ethane these peroxy radicals will lead to increases in  $NO_2$ . We have added the following sentence regarding the sensitivity of ECHAMP to these radicals at the end of section 2.2:

"We estimate an elevated uncertainty of  $\sim$ 50% for the measurements at night as we have not investigated the sensitivity of ECHAMP to peroxy radical produced by ozonolysis and NO3 reactions."

We are reluctant to comment on the measured vs. modelled ratio during the night for two reasons: 1. The measurements (of both peroxy radicals and the crucial compound NO) had much lower signal-to-noise ratios and higher uncertainties at night, and 2. The measurement height was only 3 meters which complicates the interpretation of the data since the air was usually stagnant.

*Pg* 8, *line* 20: *Could the authors make it clear which conversion efficiencies were measured and which have been estimated.*

The sentence has been re-worded to clarify:

"The conversion efficiencies for other major RO2 radicals are estimated as 5% for CH3O2 and..."

Pg 8, line 22: The authors reference the Fuchs et al., 2011 work on RO2 interferences in FAGE instruments. ' $\alpha$ ' is very much dependent upon the specific FAGE instrument and experimental conditions used, however. Using  $\alpha$  determined using another FAGE instrument would likely bias the HO2\* model measurement comparison. The authors need to make it clear how  $\alpha$  was estimated for the RO2 species not experimentally tested with the IU-FAGE. Specifically, how was  $\alpha = 0.7$  derived in equation 5 on page 14, line 21?

The sentence on pg 5 has been edited as follows:

"The conversion efficiencies for other major  $RO_2$  are estimated as 5% for  $CH_3O_2$  and the acetyl peroxy radical ( $CH_3C(O)O_2$ ), 8% for ethyl peroxy radical ( $C_2H_5O_2$ ), and 31-55% for  $RO_2$  compounds from the OH oxidation of high-molecular-weight hydrocarbons based on comparisons to several other interference tests (Fuchs et al., 2011;Griffith et al., 2016;Lew et al., 2018)."

The value of 0.7 for the "other" category was chosen since light alkanes, which have low values of  $\alpha$ , comprise a minor component of the OH reactivity and most other RO2 compounds have  $\alpha$  values near 0.7.

We have added the following sentences:

"The  $\alpha$  values for ECHAMP are based on the calculated yields of alkyl nitrates and alkyl nitrites as described in section 2.2. For LIF-FAGE, the  $\alpha$  value for C5H8(OH)O2 was measured and  $\alpha$  for CH3O2 and CH3C(O)O2 are based on measured yields to several similar instruments all of which have measured values less than 5%. An  $\alpha$  of 0.7 is assume for the "other" category since most alkenes have  $\alpha$  values between 0.5 and 0.9, and small alkanes, which have lower values, account for a small portion of the OH reactivity (Lew et al., in preparation)."

**Pg* 12, line 12, fig 3: Add the limit of detection of XO2 to the figure. Also make it clear in the figure caption which instrument measured $HO2+RO_2$**

The limit of detection (LOD) depends on the relative humidity and the variability in the ambient ozone concentration as described in Wood et al. 2017. To exactly determine the LOD at any given time requires operating both reaction chambers in background mode, precluding simultaneous knowledge of the exact LOD and the ambient concentrations. Rather than add an estimated limit of detection to the figure, we have added the following text to the caption:

"The sum of [RO2] and [HO2] was measured by the ECHAMP instrument, with a detection limit typically between 1 and 2 ppt (signal-to-noise ratio of two)."

Pg 12, section 3.3: The authors acknowledge that comparing the 30 min averaged ECHAMP measurements to a single FAGE measurement made during the 30 minute bin is not ideal. I worry that this approach could introduce bias into the comparison, given that the peroxy radical concentrations will generally be increasing throughout the morning hours and then decreasing during the afternoon and evening. Are the FAGE HO2\* measurements made at the midpoint of each 30 minute bin? Does the gradient XO2 vs HO2\* vary if the FAGE measurement falls at the start of a 30 minute bin? I think the authors need to explore the robustness of this averaging approach used for the ECHAMP data to satisfy the reader that the two measurements are comparable at the times they are taken.

We have majorly revised the paragraph below. We have also changed the averaging time used for the linear regression:

"A bi-variate linear regression of the  $XO_2$  and  $HO_2^*$  measurements between 09:00 and 22:00 yields the relationship  $[XO_2] = (1.08 \pm 0.05) [HO_2^*] - (1.4 \pm 0.3)$  ppt (Fig 6.). The regression is restricted to this window of time because of the degraded precision of the ECHAMP measurements at night due to the higher relative humidity. The  $[XO_2]/[HO_2^*]$  slopes were highest on the last two days of measurements – 24 and 25 July, with slopes of 1.25 and 1.08, respectively, or 1.5 and 1.3 after adjusting for the calibration difference. These two days were characterized by the highest mixing ratios of peroxy radicals, O3, isoprene, and the anthropogenic VOCs ethene and ethyne. The lowest  $[XO_2]/[HO_2^*]$  ratios were observed on 13 July during which a passing thunderstorm led to low concentrations during mid-day with higher values before and after the storm. The higher  $[XO_2]/[HO_2^*]$  ratios observed later in the field campaign may simply be the result of a change in sensitivity in one of the instruments. These linear are difficult to interpret, however, since the  $XO_2$  measurements are 30 minute averages and the  $HO_2^*$  measurements are 1-minute averages taken every 30 minutes. A regression of the binned data shown in Fig. 5 gives the relation  $[XO_2] = 1.0 \pm 0.14 [HO_2^*] + (1.5 \pm 1.6)$  ppt; accounting for the calibration difference gives an adjusted slope of 1.2. The [XO2]/[HO2\*] ratio using the binned data was highest between 9:45 and 10:45 (Fig. 5), but was between 0.9 and 1.1 between 14:45 and 19:15. This overall temporal trend is apparent in several days (Fig. 4). Applying a 30-min offset to the XO2 data largely removes this trend and leads to

fewer time periods when  $[XO_2]/[HO_2^*]$  was less than 1.0, but such an offset does not agree with the synchronized time-base of both measurements. The two instruments' different averaging times and precision levels preclude further assessment and conclusions regarding possible time offsets."

Page 13, line 6 - 8: the authors report the highest XO2:HO2\* ratio on days when isoprene and ethene concentrations were most elevated. This is unexpected, given the high sensitivity of FAGE to alkene-derived RO2 species. Could the authors comment on this finding?

We have edited that sentence:

"These two days were characterized by the highest mixing ratios of  $O_3$ , isoprene, and the anthropogenic VOCs ethene and ethyne. The high [XO2]/[HO2\*] ratios observed those days may simply be the result of a change in sensitivity in one of the instruments."

Page 13, line 10: The data in figure 5 has already been binned and then averaged over 9 days. Does the linear regression on the figure 5 data provide a reduced uncertainty relative to the data presented in figure 6? Errors on the fit should be included. I may have misunderstood, but don't both linear regressions use the same data (just one if further averaged into a diurnal)? Does the change in the regression slope as the data is averaged further suggest that the binning approach is biasing the correlation?

The binning is useful because there are occasional gaps in the time series (e.g., the morning of 14 July). Without the binning, the morning data is slightly "underrepresented" because of that gap. We have changed the caption as follows (including the fit errors):

"Figure 6. Correlation of ambient [XO2] measured by ECHAMP with [HO2\*] measured by IU-LIF-FAGE. The linear fit is for data between 09:00 and 22:00, indicated by the points with green circles. The equation of the fit is  $[XO_2] = (1.08 \pm 0.05) [HO_2^*] - (1.4 \pm 0.3) \text{ ppt.}$ "

Page 13, line 23: Although I appreciate that the authors do not know the reason why the measurements diverge on the 22nd, the possible explanation 'a transient interference in the HO2\* measurement when sampling ambient air..' is rather vague. Could the authors elaborate on what they think this transient interference may be or what it may be related to?

We agree that the explanation of a "transient interference" is vague, but feel that any possible reason offered at this point would be too speculative. We note that since  $HO_2^*$  is measured as OH after conversion by reaction with NO, any interference in the OH measurement would affect the  $HO_2^*$  measurements as well.

Pg 13, line 25 – Pg 14, line 4: I suggest moving this paragraph to the start of section 3.3. It is important that the  $\alpha$  of two instruments to different RO2, and how the ratio is expected to change as ambient RO2 types vary, is set out at the beginning of this section.

We agree and have made that paragraph the 2nd paragraph of section 3.3 in the revision.

Section 3.3: in general, there is a lot to consider when comparing HO2\* and XO2 measured and modelled. The ratio varies with RO2 type present and calibration differences also need to be considered. A table detailing the measured HO2\*, XO2 and XO2:HO2\* and the 4 modelled HO2\*, XO2 and XO2:HO2\* on the individual days and campaign average would help to clarify the text.

We hope that the majorly revised paragraph quoted earlier (starting with "A bi-variate linear regression...") has clarified these issues. Furthermore, the results from the 4 models are shown in the SI.

*Figure 4: The caption on the figure is obscuring the top x-axis* This has been fixed in the revision.

Figure 5: There does not seem to be a measured ratio for each 30 min point? Between the hours of 4 - 8, there are only 3 points?

Because of the low signal-to-noise ratios for the nighttime measurements (especially by ECHAMP), the ratio of the measured  $XO_2/[HO_2^*]$  varies greatly at night, from 0.3 to 2.1, and so some of those points were off-scale (the graph's axis was from 0.8 to 1.8). For the revision, we only show the ratio for time periods between 08:00 and 22:00, with the following revised caption:

"...The upper plot shows the  $[XO_2]/[HO_2^*]$  ratio - both measured by the two instruments and modeled using the MCM 3.2 chemical mechanism. The measured ratio is only shown for time periods between 08:00 and 22:00 due to the poor signal-to-noise ratios for the night-time measurements."

Figure 4 – Figure 8: It is unclear whether the ECHAMP data has been corrected for the calibration comparison or not? This should be clear in each figure caption

We have added the following sentence to section 3.3 to clarify:

"No adjustments have been made to either of the datasets in Fig. 4 (or any other figures) to account for the calibration difference."

Reviewer #2's comments:

The paper describes the measurement of peroxy radicals (HO2, RO2) with two different techniques. The LIF-FAGE technique by Indiana University was originally designed to measure solely HO2 radicals by chemical conversion with NO to OH, which is then detected by LIF. However, different experimental studies (including a study by authors of this paper) have shown that the technique is also sensitive to specific RO2 radicals with different sensitivities when the instrument is tuned for maximum HO2-to-OH conversion efficiency. The measured quantity is called HO2\*. The new ECHAMP technique, a chemical amplifier using ethane instead of CO, is designed to measure the sum of HO2 and all RO2 species. Due to different amplifier chain lengths for different radical species, the resulting quantity XO2 is a proxy for the total peroxy radical concentration. Comparing the measurements by the two techniques sounds like comparing apples with oranges. The present paper demonstrates that such a comparison can be done in a

meaningful way, if the instruments are carefully characterized and additional information about the peroxy radical speciation is available (here from box model calculations constrained by measured trace gases). The direct comparison of the conceptually different calibration methods (photolysis of water vapor vs. photolysis of acetone) and the field comparison show that the measurement techniques yield consistent data within the specified experimental uncertainties. These findings suggest that the two described methods can also be used for meaningful tests of atmospheric chemistry models, if the measured peroxy radicals (HO2\*, XO2) are appropriately simulated by the model by taking RO2-specific weighting factors of the instruments into account. This requirement should be explicitly stated in the conclusions.

Furthermore, recent progress in the measurement of HO2 by LIF-FAGE instruments should be mentioned. It has been shown that the interference by RO2 can be avoided by reducing the concentration of NO that is used for conversion to OH (e.g. Fuchs et al., 2011; Whalley et al., 2013; Feiner et al., 2016; Tan et al., 2017).

This section has been re-written as follows:

"The sensitivity of the LIF-FAGE technique to each type of organic peroxy radical varies with the amount of NO added for the conversion and is instrument-dependent but in general is highest (up to ~90%) for  $\beta$ -hydroxy peroxy radicals derived from alkenes and lowest for those derived from small alkanes (Fuchs et al., 2011;Lew et al., 2018;Whalley et al., 2013). This RO2 interference can be greatly reduced by use of lower NO concentrations or reaction times, yielding conversion efficiencies for isoprene-RO2 lower than 20% (Feiner et al., 2016;Fuchs et al., 2011;Tan et al., 2017;Whalley et al., 2013)."

Overall, the paper is thoroughly and well written. It is suitable for ACP, but could have been submitted to AMT as well. The authors and editor should consider whether the paper should appear as a "Technical note" in ACP. I recommend publication after the following minor comments have been addressed.

We agree that the paper could have been suitable for AMT as well. We chose to submit to ACP because we think that the comparison of the measured concentrations with those by the models provided information beyond that of an instrument assessment and provided information on our community's understanding of HOx chemistry in low-NOx, high biogenic VOC environments, which has historically been problematic.

(1) Introduction: as the topic of the paper is an instrumental intercomparison, I suggest to provide a more complete list of previous intercomparisons. For instance, Mount et al. (JGR vol.102, no.D5, p6437, 1997), Zenker et al. (JGR vol 103, no Dll, p13,615, 1998), Ren et al. (JGR vol 108, no D19, 4605, 2003), Fuchs et al. (AMT 5, 1611–1626, 2012), Onel et al. (AMT 10, 4877–4894, 2017), Sanchez et al. (Atmos. Env. 174, 227–236, 2018).

We have added the suggested references.

(2) In the experimental section, the authors point out that the use of ethane instead of CO offers advantages. Safer operation is obviously a plus. However, I don't understand

why the choice of ethane reduces the sensitivity on relative humidity. Is this due to the reduced chain length? Is there evidence for water influence on the OH+CO reaction? To my knowledge, the water effect has been attributed to the reaction HO2+NO (e.g., Mihele et al. 1999, Butkovskaya et al., 2007). Why is the amplification factor lower, if ethane is used? Another advantage of ethane could be mentioned, although it is probably not relevant in a forest environment. Ethane avoids possible interferences from ClOx, which can lead to amplification in CO/NO systems (Perner et al., J. Atmos. Chem. 34, 9, 1999).

We have added the following text to briefly clarify the important issue of RH-dependence:

"The cause of the RH-dependence of the CO-based amplification chemistry is the RH-dependence of the main radical termination step: the reaction of HO2 with NO to form HNO3 (Butkovskaya et al., 2007;Butkovskaya et al., 2005;Butkovskaya et al., 2009;Mihele et al., 1999;Reichert et al., 2003), with a smaller contribution from the RH-dependent wall losses of HO2. These two RH-dependent radical termination steps affect the ethane-based amplification chemistry as well, but the most important terminations steps are from the formation of ethyl nitrite and ethyl nitrate – neither of which depends on relative humidity."

(3) Page 9: "For this project, [O3] was instead quantified by the ECHAMP CAPS NO2 sensors after conversion to NO2 by reaction with excess NO". A few details should be explained: is the flow in the calibrator laminar or turbulent? Where is the NO added (upstream, downstream of the calibrator)? Is the NO2 measured after it has been passed through the FAGE cell or is it measured in the air that bypasses the inlet of the FAGE cell? How much NO is added and how large is the resulting NO2 mixing ratio?

We have edited the following section in order to provide more information on this quantification:

"For this project,  $[O_3]$  was instead quantified by the ECHAMP CAPS NO2 sensors after conversion to NO2 by reaction with excess NO. This was accomplished by having the IU calibration source overflow the ECHAMP inlet. ECHAMP was operated without the ethane flowing, so that each reaction channel sampled 1 LPM of air from the cal source into which 80 sccm of 21 ppm NO was added. This resulted in a diluted concentration of 1.7 ppm NO, which is high enough to react with 99% of the O3 formed during the transit from the inlet to the CAPS detectors. This produces a very precise measurement of the sum of  $[O_3]$  and  $[NO_2]$  (1 $\sigma$  precision of 22 ppt for 10 second averages). The accuracy of this ozone determination is thus ultimately traceable to the CAPS NO2 calibration (see SI). Typical  $[O_3]$  values measured were between 0.4 and 2.0 ppb."

(4) Page 9, line 18: is the water vapor correction based on laboratory characterization of the LIF-FAGE instrument, or on theoretical calculations using published data for the OH fluorescence lifetime and cross sections for quenching?

It is based on laboratory characterizations. The new sentence:

"The sensitivity of the instrument is corrected for fluorescence quenching by water vapor as per laboratory characterization"

(5) Model constraints: was atmospheric CO measured? Which formaldehyde data were used (GC-FID or DNPH)?

CO was not measured but was estimated based on published emission ratios of CO with benzene. Formaldehyde was only measured using the DNPH cartridges.

The relevant sentences were edited as follows:

"...cartridges to measure carbonyls, including formaldehyde (which was not measured by the GC-FID system), acetaldehyde and..."

and

"Measured VOC concentrations (every 90 min) were interpolated on to this 30 min time resolution. Carbon monoxide was not measured but instead estimated based on emission ratios of CO with benzene (Warneke et al., 2007)."

*I see large gaps in the measured time series of NO in the first half of the campaign. Was NO (when available) used as a constraint, or was NO calculated by the model using NO2 as a constraint?*

The comparison to the models is heavily focused on the three days when there measurements of NO,  $XO_2$  (by ECHAMP) and HO2\* (by LIF-FAGE) as described in the text:

"Due to gaps in the NO data because of problems with the Thermo chemiluminescence sensor, there are only three days for which we have model results and measured peroxy radical concentrations by both ECHAMP and LIF-FAGE – on the 16th, 22nd, and 24th of July. The model was run for these three days, and also a diurnal profile for the entire campaign was run using diurnal average concentrations of constrained species."

The box model was constrained with 30 minute average mixing ratios. As peroxy radicals show a strong non-linear dependence on NO, using 30 minute average values as constraint can lead to systematic bias in the model results. I would like to see the model results that are averaged to 30 minutes after the model has been run at the much higher time resolution of the NOx measurements.

The time resolution of the model is limited by the 90-minute frequency of the VOC measurements which we have interpolated to values every 30-minutes. Thus we are unable to run the model at higher time resolution.

(6) Figure 4 - 6: Is it meaningful to adjust the result of the linear regression for the calibration difference (section 3.1)? This would only make sense, if the calibration would be done for the same peroxy radical speciation as encountered during the measurement days in the field.

We have intentionally included in the text both the "raw" regression/ratio results and those corrected for the calibration difference. Since both ECHAMP and LIF-FAGE are both sensitive (high  $\alpha$ ) to HO2 and isoprene RO2, we do think that "correcting" the comparisons for the 20% calibration difference helps to frame the discussion of the differences between the two measurements.

(7) Figure 1: what is the scale of the map? The caption has been updated to address this: "The arrow represents a distance of 1 km."

(8) Figure 3: what is causing the noise and spikes on the NO data? Is it measurement precision or atmospheric variability from nearby NO sources?

The data shown in figure 3 are the 5-minute averaged NO concentrations which have a  $1\sigma$  precision of approximately 100 ppt. The "spikes" in the figure are actually of 15 to 60 minute duration, and thus are from atmospheric variability (mostly during the early morning).

(9) Figure 3 and 4: vertical dotted lines = midnight ?

Yes. We have updated the figure captions to clarify:

"The vertical grid lines indicate midnight for odd-numbered days, in local time."

(10) Figure 5: the shown error bars (1sigma precisions) seem too large compared to the variability of the shown data

In the caption for figure 5 we had erroneously described the error bars as indicative of the  $1\sigma$  precision of the measurements when they actually just describe the distribution of the measured concentrations. We have changed that sentence in the caption to the following:

[revised manuscript text omitted]

5 Shuvashish Kundu1\*, Benjamin L. Deming1\*\*, Michelle M. Lew2, Brandon P. Bottorff2, Pamela Rickly3\*\*\*, Philip S. Stevens2,3, Sebastien Dusanter4, Sofia Sklaveniti3,4, Thierry Leonardis4, Nadine Locoge4, and Ezra C. Wood5

1Department of Chemistry, University of Massachusetts, Amherst MA, 01003, United States 2Department of Chemistry, Indiana University, Bloomington IN 47405, United States

3School of Public and Environmental Affairs, Indiana University, Bloomington, IN 47405, United States 4IMT Lille Douai, Université Lille, Département Sciences de l'Atmosphère et Génie de l'Environnement (SAGE), F-59000 Lille, France 5Department of Chemistry, Drexel University, Philadelphia PA, 19104, United States

15 \* now at Momentive Performance Materials, Inc., Tarrytown, NY 10591, United States \*\* now at Department of Chemistry, University of Colorado, Boulder CO 80309, United States \*\*\* now at Cooperative Institute for Research in Environmental Sciences, University of Colorado, Boulder, CO 80309, USA and Chemical Sciences Division, Earth System Research Laboratory, National Oceanic and Atmospheric Administration, Boulder, CO 80305, USA

Correspondence to: Ezra Wood (Ezra.Wood@drexel.edu)

20

Abstract. Peroxy radicals were measured in a mixed deciduous forest atmosphere in Bloomington, Indiana, USA, during the Indiana Radical, Reactivity and Ozone Production Intercomparison (IRRONIC) during the summer of 2015. Total peroxy radicals ( $[XO_2] \equiv [HO_2] + \Sigma[RO_2]$ ) were measured by a newly developed technique involving nitric oxide (NO) – ethane (C2H6) chemical amplification followed by NO2 detection by cavity attenuated phase shift spectroscopy (hereinafter referred

- 5 to as ECHAMP). The sum of hydroperoxy radicals (HO2) and a portion of organic peroxy radicals ([HO2\*] = [HO2] +  $\Sigma \alpha_i$ [RiO2], 0< $\alpha$ <1) was measured by the Indiana University Laser-Induced Fluorescence / Fluorescence Assay by Gas Expansion instrument (LIF-FAGE). Additional collocated measurements include concentrations of NO, NO2, O3, and a wide range of volatile organic compounds (VOCs); and meteorological parameters. XO2 concentrations measured by ECHAMP peaked between 13:00 to 16:00 local time, with campaign average concentrations of 41 ± 15 ppt (1 $\sigma$ ) at 14:00. Daytime
- 10 concentrations of isoprene averaged  $3.6 \pm 1.9$  ppb (1 $\sigma$ ) whereas average concentrations of NOx ([NO] + [NO2]) and toluene were 1.2 ppb and 0.1 ppb, respectively, indicating a low impact from anthropogenic emissions at this site.

We compared ambient measurements from both instruments and conducted a calibration source comparison. For the calibration comparison, the ECHAMP instrument, which is primarily calibrated with an acetone photolysis method, sampled the output of the LIF-FAGE calibration source which is based on the water vapor photolysis method and, for these

15 comparisons, generated a 50-50% mixture of HO2 and either butane or isoprene-derived RO2. A bivariate fit of the data yields the relation  $[XO_2]_{ECHAMP} = (0.88 \pm 0.02) ([HO_2] + [RO_2])_{IU_cal} + (6.6 \pm 4.5) \text{ ppt}$  This level of agreement is within the combined analytical uncertainties for the two instruments' calibration methods.

A linear fit of the daytime (09:00 – 22:00) 30-minute averaged [XO2] ambient data with the 1-minute averaged [HO2\*] data (one point per 30 minutes) yields the relation [XO2] = (1.08 ± 0.05) [HO2\*] – (1.4 ± 0.3), Day to day variability
in the [XO2]/[HO2\*] ratio was observed. The lowest [XO2]/[HO2\*] ratios between 13:00 and 16:00 were 0.8 on 13 and 18 July, whereas the highest ratios of 1.1 to 1.3 were observed on 24 and 25 July – the same two days on which the highest concentrations of isoprene and ozone were observed. Although the exact composition of the peroxy radicals during IRRONIC is not known, 0-dimensional photochemical modeling of the IRRONIC dataset using the RACM2, RACM2-LIM1, MCM 3.2, and MCM 3.3.1 chemical mechanisms all predict afternoon [XO2]/[HO2\*] ratios of between 1.2 to 1.5.
Differences between the observed ambient [XO2]/[HO2\*] ratio and that predicted with the 0-D modeling can be attributed to deficiencies in the model errors in the two measurement techniques, or both. Time periods in which the ambient ratio was less than one are definitely caused by measurement errors (including calibration differences) as such ratios are not physically meaningful. Although these comparison results are encouraging and demonstrate the viability of using the new ECHAMP technique for field measurements of peroxy radicals, further research investigating the overall accuracy of the measurements

30 and possible interferences from both methods is warranted.

[revised manuscript text omitted]

where [O3] is the concentration of ozone generated by the photolysis of O2; σH2O and σO2 are the absorption cross sections of H2O and O2 at 184.9 nm, respectively; and φH2O and φO2 are the photolysis quantum yields, both equal to two (Washida et al., 1971). A value of 7.14 × 10-20 cm2 molecule-1 (base e) was used for σH2O (Cantrell et al., 1997;Hofzumahaus et al., 1997;Lanzendorf et al., 1997). The effective value of σO2 depends on the O2 optical depth and the operating conditions of the mercury lamp and was determined to be 1.20 × 10-20 cm2 molecule-1 (Dusanter et al., 2008;Lanzendorf et al., 1997). The water vapor mixing ratio was measured by IR absorption spectrometry using a LI-COR 6262 monitor. Ordinarily the ozone mixing ratio is determined using a calibrated photodiode installed in the calibrator (Griffith et al., 2013). The conversion factor (calibration) that converts the photodiode reading to an O3 mixing ratio is determined from separate experiments in which a range of O3 concentrations produced by the calibrator are measured with a UV-absorption O3 sensor. For this

project, [O3] was instead quantified by the ECHAMP CAPS NO2 sensors after conversion to NO2 
[revised manuscript text omitted]

**Moved (insertion) [1]**

**Field Code Changed**

| 1            | Deleted: all                   |
|--------------|--------------------------------|
| 1            | Deleted: measurements          |
| Ι            | Deleted: 0.85                  |
|              | Deleted:                       |
| $\mathbb{Z}$ | Deleted: + 3.3          |
| $\mathbb{N}$ | Deleted: , $R^2 = 0.67$ |
|              |                                |

**Deleted:** Adjusting this for the calibration difference described section 3.1, i.e. increasing the  $XO_2$  values by a factor of 1.2 yield slope of 1.02. If the y-intercept is constrained to zero, which is n necessarily appropriate, the slope increases to 0.96, or 1.15 after adjusting for the calibration difference.

| Deleted: individual days with the highest             |
|-------------------------------------------------------|
| Deleted: s were                                       |
| Deleted: and 25                                       |
| Deleted: , with slopes of 1.25 and 1.08, respectively |
| Deleted: (                                            |
| Deleted: and 1.3                                      |
| Deleted: )                                            |
| Deleted: These two days                               |
| Deleted: .                                            |
| Deleted: is                                           |
| Deleted: fit of all the data is                |
| Deleted: 0.98                                         |
| Deleted: 7                                            |
| Deleted: 18                                           |
| Deleted:                                              |

Between 15:00 and 17:00, the HO2\* measurements increased from 50 to 70 pptv and decreased back to 50 pptv while the  $XO_2$  measurements were relatively invariant at 40 pptv. Ignoring the difference between the average mixing ratios, this difference in the temporal profile of the two instruments' measurements result could only be "real" if there were changes in the peroxy radical relative composition on this two-hour time scale, e.g. a simultaneous increase in HO2 and a decrease in

- 5 alkyl peroxy radicals, such that [HO2\*] actually did increase while the mixing ratio of total peroxy radicals was almost constant. Measurements of VOC composition and NOx do not support such a fast change in peroxy radical composition, suggesting that these observations were more likely the result of an instrumental issue. Currently we are unable to identify the exact cause of this observation, but possible explanations are a transient interference in the HO2\* measurement when sampling ambient air or a change in the sensitivity of the ECHAMP measurements.
- Because the composition of the peroxy radicals during IRRONIC is not exactly known, we examine the predicted speciation generated by zero-dimensional photochemical modeling of the IRRONIC dataset using two versions of the Regional Atmospheric Chemistry Mechanism (RACM2 and RACM2-LIM1) and the Master Chemical Mechanism (MCM 3.2 and 3.3.1). A full comparison of the modeled and measured concentrations is beyond the scope of this paper; we use these model outputs mainly to inform the discussion of the relative speciation of total peroxy radicals and its relation to the expected and measured [XO2]/[HO2\*] ratio. A fuller description of the photochemistry at this site, including OH reactivity
  - measurements, will be described in a companion paper (Lew et al, in preparation).

The accuracy of the model results is, of course, subject to how comprehensive and accurate the supporting measurements and underlying chemical mechanisms are, but nonetheless help to frame the interpretation of the two instruments' measurements. Due to gaps in the NO data because of problems with the Thermo chemiluminescence sensor,

20 there are only three days for which we have model results and measured peroxy radical concentrations by both ECHAMP and LIF-FAGE – on the 16th, 22nd, and 24th of July. The model was run for these three days, and also a diurnal profile for the entire campaign was run using diurnal average concentrations of constrained species. From these model results we calculate the expected values measured by ECHAMP and LIF-FAGE based on each instrument's relevant values for α:

25 ECHAMP  $[XO_2]_{EXPECTED} = [HO_2] + 0.9([CH_3O_2]) + 0.92([C_3H_8(OH)O_2]) + 0.9([CH_3C(O)O_2]) + 0.9(Other)$  (4)

 $LIF-FAGE [HO_2^*]_{EXPECTED} = [HO_2] + 0.05([CH_3O_2]) + 0.83([C_5H_8(OH)O_2]) + 0.05([CH_3C(O)O_2]) + 0.7(Other)$ (5)

The "Other" category includes all types of peroxy radicals, e.g., from monoterpenes, methyl vinyl ketone, ethene, etc. The  $\alpha$ 30 values for ECHAMP are based on the calculated yields of alkyl nitrates and alkyl nitrites as described in section 2.2. For LIF-FAGE, the  $\alpha$  value for C5H8(OH)O2 was measured and  $\alpha$  for CH3O2 and CH3C(O)O2 are based on measured yields from several similar instruments, all of which have measured values less than 5%. An  $\alpha$  of 0.7 is assumed for the "other" category since most alkenes have  $\alpha$  values between 0.5 and 0.9, and small alkanes, which have lower values, account for a small portion of the OH reactivity (Lew et al., in preparation). Moved up [1]: The "true" [XO2]/[HO2\*] ratio, i.e., the ratio th would be produced by the two instruments' measurements if the were calibrated to the same source and operated exactly as expect without any uncharacterized interferences or losses, depends on composition of the peroxy radicals. As described in Section 2 (Experimental Methods), for both ECHAMP and LIF-FAGE, the sensitivity of the instrument to individual RO2 compounds deper on the R-group and is characterized by the parameter "a", which the instrument's sensitivity to each RO2 relative to its sensitivity HO2. For ECHAMP α is determined largely by the fraction of R that is converted to alkyl nitrates (RONO2) and alkyl nitrites (RONO) following reaction with NO at atmospheric pressure. For LIF-FAGE, a is mostly determined by how quickly each RO2 is converted sequentially to HO2 and then OH following reaction w NO after the expansion of the sampled gas into the low-pressure region of the instrument (Fuchs et al., 2011;Lew et al., 2018). Ai which CH3O2, CH3C(O)O2, and small (<C5) alkyl peroxy radical have a large contribution to the total peroxy radical concentration would thus produce a relatively high [XO2]/[HO2\*] value, since ECHAMP is sensitive to those peroxy radicals ( $\alpha > 0.9$ ) whereas t LIF-FAGE HO2\* measurement is not ( $\alpha < 0.1$ ). In contrast, air wi relatively high fraction of alkene-derived RO2 (e.g., isoprene per radicals), for which both ECHAMP and LIF-FAGE HO2\* a valu are near one, would be expected to lead to lower [XO2]/[HO2\*] values (i.e., closer to unity). ¶

The top portion of Fig. 5 shows the average diurnal profile for the  $[XO_2]/[HO_2*]$  ratiomodeled by MCM 3.2 and measured using all days when there were both XO2 and HO2\* measurements. Between 10:00 and 18:00 the modeled  $[XO_2]/[HO_2*]$  ratio using MCM 3.2 varied between 1.2 and 1.5, whereas the measured ratio varied between 0.9 and 1.4, with a greater amount of variability from hour to hour. Increasing the observed ratio by 20% to account for the calibration comparison (section 3.1) gives an adjusted measured ratio of between 1.1 and 1.7. The highly variable ratios during nighttime mainly reflect the lower signal to noise ratios of both instruments when peroxy radical concentrations were low (less than ~5 ppt).

Measured and MCM 3.2 modeled concentrations for 16, 22, and 24 July are shown in Fig. 8. On all three days the relative contributions from the various types of peroxy radicals are comparable. At 15:30 –when concentrations were highest
- the modeled peroxy radicals comprised 30% C5H8(OH)O2, 35% HO2, 26% CH3O2 and 7% CH3C(O)O2. The four chemical mechanisms vary little in the predicted relative speciation (SI). The [XO2]/[HO2\*] ratio modeled by MCM 3.2 between 15:00 and 16:00 is 1.4 for 16 and 22 July and 1.45 on 24 July. The measured [XO2]/[HO2\*] ratio is close to unity on 16 and 22 July, and between 1.2 and 1.5 on 24 July. Increasing these measured ratios by 20% to account for the calibration comparison produces adjusted measured [XO2]/[HO2\*] ratios of 1.2 on 16 and 22 July and 1.4 to 1.8 on 24 July. After accounting for the 20% calibration difference, the modeled and measured ratios agree to within the experimental and model uncertainties.

Although all four chemical mechanisms predict a very similar relative speciation, there are variations in the absolute peroxy radical concentrations predicted. MCM 3.3.1 concentrations are very similar to those from MCM 3.2, but RACM2 and RACM2-LIM1 predict 26% and 42% higher peak concentrations, respectively. Measured [XO2] mixing ratios are 20 to 30% lower than the MCM 3.2, [XO2] on 16 and 22 July but agree more closely on 24 July (measured/modeled ratio varies from 0.8 to 1.15). The comparison between measured [HO2\*] and modeled [HO2\*] for these three days exhibits more

variability (Fig. 8), Further details can be found in the SI.

Observations of [XO2]/[HO2\*] ratios less than one were observed during parts of 13, 17, and 18 July and even after increasing by 20% to account for the calibration comparison do not seem reasonable or in some cases even possible. These observations were most likely caused by issues with one or both instruments. Two possible causes that warrant investigation

25 in subsequent field measurements are discussed below:

1. Error in the ECHAMP calibration, especially for RH values greater than 45%. Although the calibration comparison presented in section 3.1 show that the ECHAMP and LIF-FAGE instrument's calibrations agreed to within measurement uncertainties, that is not necessarily true for RH values greater than those used during those calibration tests. The highest RH

30 value during the calibration comparisons was 45%, whereas the daytime minimum RH values between 12:00 and 16:00, when measured  $[XO_2]$  and  $[HO_2^*]$  were both highest, were typically between 45% and 65% (Fig 1). Furthermore, we cannot prove that the ECHAMP calibration was invariant from day to day. We include potential sampling losses to be a part of the overall ECHAMP calibration.

| Deleted: modeled and measured                                                                                                       |
|-------------------------------------------------------------------------------------------------------------------------------------|
| Deleted: RACM2                                                                                                                      |
| Deleted: 4                                                                                                                          |
| Deleted: RACM2                                                                                                                      |
| Deleted: are                                                                                                                        |
| Deleted: 40                                                                                                                         |
| Deleted: 33                                                                                                                         |
| Deleted: 1                                                                                                                          |
| Formatted: Not Superscript/ Subscript                                                                                               |
| Formatted: Not Superscript/ Subscript                                                                                               |
| Deleted: CH 3 O 2                                                                                             |
| Deleted: ,                                                                                                                          |
| Deleted: 4                                                                                                                          |
| Formatted: Not Superscript/ Subscript                                                                                               |
| Deleted: O 2 , and 7% "Other", resulting in an expected (mode [XO 2 ]/[HO 2 *] ratio of 1.3 |
| Deleted: results                                                                                                                    |
| Deleted: ~40                                                                                                                        |
| Deleted: modeled                                                                                                                    |
| Deleted: within                                                                                                                     |
| Deleted: 10%                                                                                                                        |
| Formatted: Highlight                                                                                                                |
| Deleted: ), but except for a few short time periods modeled                                                                  |

 $[HO_2^*]$  is greater than measured  $[HO_2^*]$ . Choice of chemical mechanism

**Deleted:** Although all four chemical mechanisms predict a versimilar relative speciation, there are variations in the absolute peradical concentrations predicted/dose lead to differences in both tabsolute concentrations and relative speciation. Compared MCM 3.3.1 concentrations are very similar to those from MCM 3.2, bu RACM2, and RACM2-LIM1 predict leads to 2260% and 42% higher peak concentrations, respectively though the relative speciation is very similar. Concentrations produced by MCM v3 are similarly 20% higher than RACM2, though CH5C(O)O2 accc for 9% of the total peroxy radical concentration in the afternoon compared to 4% for RACM2. MCM v3.3.1 leads to the highest overall concentrations (~30% higher than RACM2), a 12% contribution from CH3C(O)O2, and only a 19% contribution fror C5H8(OH)O2.

Deleted: In summary, the [X02]/[H02\*] ratios modeled by the four chemical mechanisms vary from 1.3 (RACM2 and RACM2 LIM1) to 1.4 (MCM v. 32) to 1.5 (MCM v. 331) between 15:00 16:00, whereas the measured ratios varied from 0.8 to 1.3 depen on the day, with an average value of 1.0. After accounting for th 20% calibration difference, the modeled and measured ratios agr to within the experimental and model uncertainties. That both th measured [X02] and [H02\*] concentrations, however, are typica only 50% of the values predicted by MCM v3.3.1 does suggest t 2. Interferences in the LIF-FAGE measurement. The comparison of high temporal resolution in Fig. 7 revealed differences in the temporal profile of the LIF-FAGE and ECHAMP sensor. If these were caused by an interference in the LIF-FAGE measurement when sampling ambient air, then it would follow that the two instruments would agree when sampling a calibration source but differ when sampling ambient air.

5

As discussed earlier, the RH-dependence of the sensitivity of chemical amplifiers has recently been questioned (Sommariva et al., 2011). Had we ignored the RH dependence for ECHAMP's amplification factor and simply used the value under dry conditions, the daytime  $XO_2$  values would have been roughly 50% lower than those presented in this paper, leading to unrealistically low  $[XO_2]/[HO_2*]$  ratios of ~0.5.

**10 4. Conclusions**

The results of this comparison of the IU calibration source and the ambient measurements of peroxy radicals by ECHAMP and LIF-FAGE provide encouraging first results that the newly developed ECHAMP method can be used for ambient measurements of total peroxy radicals. The ECHAMP measurements, based on the acetone photolysis method, and the IU water vapor photolysis calibration source agreed within 12%, within the experimental uncertainties. The measured mixing

15 ratios of XO2 and HO2\* were usually lower than the concentrations predicted by the RACM2, RACM2-LIM1, MCM v. 3.2, and MCM v. 3.3.1 chemical mechanisms. The measured [XO2]/[HO2\*] ratios usually differed from the ratios predicted by zero-dimensional photochemical modeling by less than the combined measurement and modeling uncertainties, though the lowest ratios observed (0.8) are not physically meaningful and therefore must be due to measurement errors.

An attribute of these comparison exercises is that the two instruments operate on very different measurement principles and the calibration methods differ greatly. Although the calibration comparison was favorable, due to the time required to conduct successful calibrations with the acetone photolysis method and its overall inconvenience (Wood and Charest, 2014) we have discontinued its use. For subsequent field measurements we have used the water vapor photolysis method and another method based on methyl iodide photolysis (Anderson et al., 2019;Clemitshaw et al., 1997;Liu and Zhang, 2014), All three calibration methods do indicate that a humidity-dependent calibration must be used for both CObased and ethane-based chemical amplifiers.

**Data Availability**

Data are available upon request from the corresponding author (Ezra.Wood@drexel.edu)

**Author contributions.**

5

EW and PS designed the research project. SK, BD, and EW were responsible for the ECHAMP measurements and supporting measurements of NO, NO2, and ML, BB, PR, and PS were responsible for the LIF-FAGE measurements and photochemical modeling. SD, SS, TL, and NL were responsible for the measurements of VOCs. SK and EW conducted the analysis and wrote the paper with feedback from all co-authors.

Competing Interests. The authors declare that they have no conflicts of interest.

Acknowledgements. This study was supported by the National Science Foundation via grant NSF AGS-1443842 to the
 University of Massachusetts, NSF grant AGS-1719918 to Drexel University, and NSF grant AGS-1440834 to Indiana University. This work was also supported by grants from the Regional Council Nord–Pas-de-Calais through the MESFOZAT project, as well as the French National Research Agency (ANR–11–LABX–0005–01) and the European Regional Development Fund (ERDF) through the CaPPA (Chemical and Physical Properties of the Atmosphere) project. We are grateful to J. Flynn and B. Lefer of the University of Houston for the spectroradiometer data\_used for the chemical modelling.

**Deleted: ¶**

---

## Editor Decision (ED1)

Both reviewers have recommended publication of the manuscript subject to minor changes. However, the responses to these comments are in a few instances not sufficient. Please address the comments identified below more thoroughly prior to publication. The editor's comments to which the authors should respond are indicated in blue.

*Page 13, line 10: The data in figure 5 has already been binned and then averaged over 9 days. Does the linear regression on the figure 5 data provide a reduced uncertainty relative to the data presented in figure 6? Errors on the fit should be included. I may have misunderstood, but don't both linear regressions use the same data (just one if further averaged into a diurnal)? Does the change in the regression slope as the data is averaged further suggest that the binning approach is biasing the correlation?*

The binning is useful because there are occasional gaps in the time series (e.g., the morning of 14 July). Without the binning, the morning data is slightly "underrepresented" because of that gap. We have changed the caption as follows (including the fit errors):

"Figure 6. Correlation of ambient $[XO_2]$ measured by ECHAMP with $[HO_2^*]$ measured by IU-LIF-FAGE. The linear fit is for data between 09:00 and 22:00, indicated by the points with green circles. The equation of the fit is $[XO_2] = (1.08 \pm 0.05)\,[HO_2^*] - (1.4 \pm 0.3)$ ppt."

The reviewer asked several questions in this comment, but the response does not address the last, highlighted one above. The changes to the manuscript should include a response to this question.

*Page 13, line 23: Although I appreciate that the authors do not know the reason why the measurements diverge on the 22nd, the possible explanation 'a transient interference in the HO2\* measurement when sampling ambient air..' is rather vague. Could the authors elaborate on what they think this transient interference may be or what it may be related to?*

We agree that the explanation of a "transient interference" is vague, but feel that any possible reason offered at this point would be too speculative. We note that since $HO_2^*$ is measured as OH after conversion by reaction with NO, any interference in the OH measurement would affect the $HO_2^*$ measurements as well.

It is reasonable to state that the authors do not know the reason for the divergence of the measurements over a short period. The response does not make sense, however, since it first speculates that there is a "transient interference" in one measurement (without assigning a mechanism), but that any other explanation would be speculative (i.e, first speculates, then declines changes on the grounds that it would be speculative). If the authors are not able to follow the reviewer's suggestion as to the nature of the interference, then the speculation regarding the transient interference should simply be removed. The divergence can be noted, and the authors can also state that the reason for the divergence is not known.

*Section 3.3: in general, there is a lot to consider when comparing HO2\* and
XO2 measured and modelled. The ratio varies with RO2 type present and calibration differences also
need to be considered. A table detailing the measured HO2\*, XO2
and XO2:HO2\* and the 4 modelled HO2\*, XO2 and XO2:HO2\* on the individual days and campaign
average would help to clarify the text.*

We hope that the majorly revised paragraph quoted earlier (starting with "A bi-variate linear
regression...") has clarified these issues. Furthermore, the results from the 4 models are shown in the SI.

Can the authors provide the requested table, highlighted above? The comment has not been
addressed.

*The paper describes the measurement of peroxy radicals (HO2, RO2) with two different techniques. The
LIF-FAGE technique by Indiana University was originally designed to measure solely HO2 radicals by
chemical conversion with NO to OH, which is then detected by LIF. However, different experimental
studies (including a study by authors of this paper) have shown that the technique is also sensitive to
specific RO2 radicals with different sensitivities when the instrument is tuned for maximum HO2-to-OH
conversion efficiency. The measured quantity is called HO2\*. The new ECHAMP technique, a chemical
amplifier using ethane instead of CO, is designed to measure the sum of HO2 and all RO2 species. Due to
different amplifier chain lengths for different radical species, the resulting quantity XO2 is a proxy for the
total peroxy radical concentration. Comparing the measurements by the two techniques sounds like
comparing apples with oranges. The present paper demonstrates that such a comparison can be done in a*

*meaningful way, if the instruments are carefully characterized and additional information about the
peroxy radical speciation is available (here from box model calculations constrained by measured trace
gases). The direct comparison of the conceptually different calibration methods (photolysis of water
vapor vs. photolysis of acetone) and the field comparison show that the measurement techniques yield
consistent data within the specified experimental uncertainties. These findings suggest that the two
described methods can also be used for meaningful tests of atmospheric chemistry models, if the
measured peroxy radicals (HO2\*, XO2) are appropriately simulated by the model by taking RO2-specific
weighting factors of the instruments into account. This requirement should be explicitly stated in the
conclusions.*

*Furthermore, recent progress in the measurement of HO2 by LIF-FAGE instruments should be
mentioned. It has been shown that the interference by RO2 can be avoided by reducing the concentration
of NO that is used for conversion to OH (e.g. Fuchs et al., 2011; Whalley
et al., 2013; Feiner et al., 2016; Tan et al., 2017).*

This section has been re-written as follows:

*"The sensitivity of the LIF-FAGE technique to each type of organic peroxy radical varies with the
amount of NO added for the conversion and is instrument-dependent but in general is highest (up to
~90%) for β–hydroxy peroxy radicals derived from alkenes and lowest for those derived from small
alkanes (Fuchs et al., 2011;Lew et al., 2018;Whalley et al., 2013). This $RO_2$ interference can be greatly
reduced by use of lower NO concentrations or reaction times, yielding conversion efficiencies for
isoprene-$RO_2$ lower than 20% (Feiner et al., 2016;Fuchs et al., 2011;Tan et al., 2017;Whalley et al.,
2013)."*

It is not clear that the change addresses the comment highlighted by the reviewer above. An explicit statement that the comparison between the two techniques is meaningful if an appropriate model simulation demonstrates them to be comparable should be included. Alternatively, if the authors disagree with the comment, they should state their reasoning.

*(2) In the experimental section, the authors point out that the use of ethane instead of CO offers advantages. Safer operation is obviously a plus. However, I don't understand*

*why the choice of ethane reduces the sensitivity on relative humidity. Is this due to the reduced chain length? Is there evidence for water influence on the OH+CO reaction? To my knowledge, the water effect has been attributed to the reaction HO2+NO (e.g., Mihele et al. 1999, Butkovskaya et al., 2007). Why is the amplification factor lower, if ethane is used? Another advantage of ethane could be mentioned, although it is probably not relevant in a forest environment. Ethane avoids possible interferences from ClOx, which can lead to amplification in CO/NO systems (Perner et al., J. Atmos. Chem. 34, 9, 1999).*

We have added the following text to briefly clarify the important issue of RH-dependence:

The cause of the RH-dependence of the CO-based amplification chemistry is the RH-dependence of the main radical termination step: the reaction of $HO_2$ with NO to form $HNO_3$ (Butkovskaya et al., 2007;Butkovskaya et al., 2005;Butkovskaya et al., 2009;Mihele et al., 1999;Reichert et al., 2003), with a smaller contribution from the RH-dependent wall losses of $HO_2$. These two RH-dependent radical termination steps affect the ethane-based amplification chemistry as well, but the most important terminations steps are from the formation of ethyl nitrite and ethyl nitrate – neither of which depends on relative humidity.

A water vapor dependence in the reaction of peroxy radicals, particularly $HO_2 + NO$, has been invoked to explain observed water vapor dependences in chemical amplifiers. Nevertheless, this is not the "main radical termination step" in such amplifiers, unless I misunderstand, but rather the one that leads to a water vapor dependence. The authors may want to consider rephrasing.

*The box model was constrained with 30 minute average mixing ratios. As peroxy radicals show a strong non-linear dependence on NO, using 30 minute average values as constraint can lead to systematic bias in the model results. I would like to see the model results that are averaged to 30 minutes after the model has been run at the much higher time resolution of the NOx measurements.*

The time resolution of the model is limited by the 90-minute frequency of the VOC measurements which we have interpolated to values every 30-minutes. Thus we are unable to run the model at higher time resolution.

This response does not make sense. If it is possible to interpolate from 90 to 30 minutes, then it is also possible to interpolate to a faster time scale. It cannot be the case that the authors are thus

"unable" to run the model at higher time resolution.  If the authors feel there is nothing to be gained in doing so, that would be an acceptable response, and the authors should make this case instead.

*(6) Figure 4 - 6: Is it meaningful to adjust the result of the linear regression for the calibration difference (section 3.1)? This would only make sense, if the calibration would be done for the same peroxy radical speciation as encountered during the measurement days in the field.*

We have intentionally included in the text both the "raw" regression/ratio results and those corrected for the calibration difference. Since both ECHAMP and LIF-FAGE are both sensitive (high $\alpha$) to $HO_2$ and isoprene $RO_2$, we do think that "correcting" the comparisons for the 20% calibration difference helps to frame the discussion of the differences between the two measurements.

It is not clear why the word "correcting" is in quotes.  Again, the response does not appear to make sense.  Either the correction is justified, or it isn't, but the justification should not include "framing the discussion."  A simpler response that simply states the justification for the correction is all that appears to be required.

---

## Author Response (AR2)

Our responses are in **bold** below.

*Page 13, line 10: The data in figure 5 has already been binned and then averaged over 9 days. Does the linear regression on the figure 5 data provide a reduced uncertainty relative to the data presented in figure 6? Errors on the fit should be included. I may have misunderstood, but don't both linear regressions use the same data (just one if further averaged into a diurnal)? Does the change in the regression slope as the data is averaged further suggest that the binning approach is biasing the correlation?*

The binning is useful because there are occasional gaps in the time series (e.g., the morning of 14 July). Without the binning, the morning data is slightly "underrepresented" because of that gap. We have changed the caption as follows (including the fit errors):

"Figure 6. Correlation of ambient [XO$_2$] measured by ECHAMP with [HO$_2$*] measured by IU-LIF FAGE. The linear fit is for data between 09:00 and 22:00, indicated by the points with green circles. The equation of the fit is [XO$_2$] = (1.08 ± 0.05) [HO$_2$*] – (1.4 ± 0.3) ppt."

The reviewer asked several questions in this comment, but the response does not address the last, highlighted one above. The changes to the manuscript should include a response to this question.

**We have edited the section as follows, discussing the point of using the binned data:**

**"These linear regressions are difficult to interpret, however, since the XO$_2$ measurements are 30 minute averages and the HO$_2$* measurements are 1-minute averages taken every 30 minutes. *Furthermore, the regression with all of the data gives equal weight to each (daytime) measurement, which due to occasional gaps in the time series (e.g., the morning of 14 July), can result in certain times of day being underrepresented.* A regression of the binned data shown in Fig. 5 gives the relation [XO$_2$] = 1.0 ± 0.14 [HO$_2^*$] + (1.5 ± 1.6) ppt; accounting for the calibration difference gives an adjusted slope of 1.2. *Using the binned data gives equal weight to each 30-minute time period (between 09:00 and 22:00).* [XO$_2$]/[HO$_2$*] ratio using…"**

**That is, the change in regression slope is not due to the binning approach biasing the correlation but rather results from the different weightings as discussed in the text above.**

*Page 13, line 23: Although I appreciate that the authors do not know the reason why the measurements diverge on the 22nd, the possible explanation 'a transient interference in the HO2\* measurement when sampling ambient air..' is rather vague. Could the authors elaborate on what they think this transient interference may be or what it may be related to?*

We agree that the explanation of a "transient interference" is vague, but feel that any possible reason offered at this point would be too speculative. We note that since HO$_2$* is measured as OH after conversion by reaction with NO, any interference in the OH measurement would affect the HO$_2$* measurements as well.

It is reasonable to state that the authors do not know the reason for the divergence of the measurements over a short period. The response does not make sense, however, since it first speculates that there is a "transient interference" in one measurement (without assigning a mechanism), but that any other explanation would be speculative (i.e, first speculates, then declines changes on the grounds that it would be speculative). If the authors are not able to follow the reviewer's suggestion as to the nature of the interference, then the speculation regarding the transient interference should simply be removed. The divergence can be noted, and the authors can also state that the reason for the divergence is not known.

**We have removed the "transient interference" text and changed that sentence as follows:**

**"Measurements of VOC composition and NOx do not support such a fast change in peroxy radical composition, suggesting that these observations were more likely the result of an instrumental issue, though we are unable to identify the cause."**

*Section 3.3: in general, there is a lot to consider when comparing HO2\* and XO2 measured and modelled. The ratio varies with RO2 type present and calibration differences also need to be considered. A table detailing the measured HO2\*, XO2 and XO2:HO2\* and the 4 modelled HO2\*, XO2 and XO2:HO2\* on the individual days and campaign average would help to clarify the text.*
We hope that the majorly revised paragraph quoted earlier (starting with "A bi-variate linear regression...") has clarified these issues. Furthermore, the results from the 4 models are shown in the SI. Can the authors provide the requested table, highlighted above? The comment has not been addressed.

**For the revision we conducted and have presented modeling using only the three days with measurements of HO$_2$\* (LIF), XO$_2$ (ECHAMP), and NO: July 16, 22, and 24. We have added the requested table to the SI (and below) that summarizes the daytime (13:00 – 18:00) concentrations for measured HO$_2$\*, XO$_2$, and their ratio, along with the same quantities from the 4 models. We have not added additional text as the existing sentence should be sufficient: "**Further details can be found in the SI." (pg 16, line 23)

**Table S1**. Summary of modeled and measured concentrations and ratios between 13:00 and 18:00.

| | 16 Jul | 22 Jul | 24 Jul |
|---|---|---|---|
| **Measured** [XO$_2$] | **28.4** | **38.9** | **58.6** |
| [HO$_2$\*] | **26.9** | **34.5** | **41.5** |
| [XO$_2$]/[HO$_2$\*] | **1.06** | **1.13** | **1.41** |
| **MCM32** [XO$_2$] | 38.1 | 44.1 | 55.2 |
| [HO$_2$\*] | 29.8 | 31.4 | 38.3 |

| | | | |
|---|---|---|---|
| [XO$_2$]/[HO$_2$*] | 1.39 | 1.41 | 1.45 |
| **MCM331** [XO$_2$] | 49.8 | 47.5 | 57.2 |
| [HO$_2$*] | 35.2 | 32.8 | 38.9 |
| [XO$_2$]/[HO$_2$*] | 1.42 | 1.46 | 1.48 |
| **RACM2** [XO$_2$] | 66.1 | 56.7 | 69.4 |
| [HO$_2$*] | 50.3 | 42.4 | 51.1 |
| [XO$_2$]/[HO$_2$*] | 1.32 | 1.34 | 1.36 |
| **RACM2-LIM1** [XO$_2$] | 81.3 | 67.4 | 79.2 |
| [HO$_2$*] | 60.3 | 49.3 | 57.5 |
| [XO$_2$]/[HO$_2$*] | 1.35 | 1.37 | 1.38 |

*These findings suggest that the two described methods can also be used for meaningful tests of atmospheric chemistry models, if the measured peroxy radicals (HO2\*, XO2) are appropriately simulated by the model by taking RO2-specific weighting factors of the instruments into account. This requirement should be explicitly stated in the conclusions.*

It is not clear that the change addresses the comment highlighted by the reviewer above. An explicit statement that the comparison between the two techniques is meaningful if an appropriate model simulation demonstrates them to be comparable should be included. Alternatively, if the authors disagree with the comment, they should state their reasoning.

**We have inserted the following sentence into the conclusion:**

**"For this type of comparison of modeled to measured peroxy radicals to be meaningful, it is crucial that the model output concentrations be weighted according to both measurement techniques' sensitivities to each class of peroxy radicals."**

*(2) In the experimental section, the authors point out that the use of ethane instead of CO offers advantages. Safer operation is obviously a plus. However, I don't understand why the choice of ethane reduces the sensitivity on relative humidity. Is this due to the reduced chain length? Is there evidence for water influence on the OH+CO reaction? To my knowledge, the water effect has been attributed to the reaction HO2+NO (e.g., Mihele et al. 1999, Butkovskaya et al., 2007). Why is the amplification factor lower, if ethane is used? Another advantage of ethane could be mentioned, although it is probably not relevant in a forest environment. Ethane avoids possible interferences from*

*ClOx, which can lead to amplification in CO/NO systems (Perner et al., J. Atmos. Chem. 34, 9, 1999).*
We have added the following text to briefly clarify the important issue of RH-dependence:
The cause of the RH-dependence of the CO-based amplification chemistry is the RH-dependence of the main radical termination step: the reaction of $HO_2$ with NO to form $HNO_3$ (Butkovskaya et al., 2007;Butkovskaya et al., 2005;Butkovskaya et al., 2009;Mihele et al., 1999;Reichert et al., 2003), with a smaller contribution from the RH-dependent wall losses of $HO_2$. These two RH-dependent radical termination steps affect the ethane-based amplification chemistry as well, but the most important terminations steps are from the formation of ethyl nitrite and ethyl nitrate – neither of which depends on relative humidity.

A water vapor dependence in the reaction of peroxy radicals, particularly $HO_2$ + NO, has been invoked to explain observed water vapor dependences in chemical amplifiers. Nevertheless, this is not the "main radical termination step" in such amplifiers, unless I misunderstand, but rather the one that leads to a water vapor dependence. The authors may want to consider rephrasing.

**Based on the results from the references quoted above (Butkovskaya et al., 2007, etc.) it does indeed appear that the RH-dependent reaction $HO_2$ + NO $\rightarrow$ $HNO_3$ is the main radical termination step!**

*The box model was constrained with 30 minute average mixing ratios. As peroxy radicals show a strong non-linear dependence on NO, using 30 minute average values as constraint can lead to systematic bias in the model results. I would like to see the model results that are averaged to 30 minutes after the model has been run at the much higher time resolution of the NOx measurements.*
The time resolution of the model is limited by the 90-minute frequency of the VOC measurements which we have interpolated to values every 30-minutes. Thus we are unable to run the model at higher time resolution.
This response does not make sense. If it is possible to interpolate from 90 to 30 minutes, then it is also possible to interpolate to a faster time scale. It cannot be the case that the authors are thus

"unable" to run the model at higher time resolution. If the authors feel there is nothing to be gained in doing so, that would be an acceptable response, and the authors should make this case instead

**While the NOx data is available on a shorter averaging time, that is not the case for the VOC measurements, which were measured every 90 minutes. We have interpolated those measurements on a 30 min time scale, but feel that it would not be particularly meaningful to conduct modeling with the faster (10-second) NOx data given how much interpolation would be required for the VOC measurements. If the results were different, it could just as easily be attributed to artifacts resulting from the high degree of interpolation done for the VOC measurements.**

*(6) Figure 4 - 6: Is it meaningful to adjust the result of the linear regression for the calibration difference (section 3.1)? This would only make sense, if the calibration would be done for the same peroxy radical speciation as encountered during the measurement days in the field.*

We have intentionally included in the text both the "raw" regression/ratio results and those corrected for the calibration difference. Since both ECHAMP and LIF-FAGE are both sensitive (high α) to $HO_2$ and isoprene $RO_2$, we do think that "correcting" the comparisons for the 20% calibration difference helps to frame the discussion of the differences between the two measurements.

It is not clear why the word "correcting" is in quotes. Again, the response does not appear to make sense. Either the correction is justified, or it isn't, but the justification should not include "framing the discussion." A simpler response that simply states the justification for the correction is all that appears to be required.

**Revised response: We have intentionally included in the text both the "raw" regression/ratio results and those corrected for the calibration difference, and argue that the correction is justified since the calibration comparison was conducted with compounds to which both ECHAMP and LIF-FAGE are sensitive (high α) – $HO_2$, butane-$RO_2$, and isoprene $RO_2$. Had the calibration comparison been conducted using a peroxy radical for which the two techniques had very different α values, for example $CH_3O_2$ for which LIF-FAGE is insensitive, then in that case we would agree that such a correction would be inappropriate.**

[revised manuscript text omitted]